

# Topic models with elements of neural networks: investigation of stability, coherence, and determining the optimal number of topics

Sergei Koltcov[1], Anton Surkov[1], Vladimir Filippov[2] and Vera Ignatenko[1]

[1] Laboratory for Social and Cognitive Informatics, National Research University Higher School of Economics, Saint-Petersburg, Russia
[2] Scientific Research Institute for Optoelectronic Instrument Engineering, Sosnovy Bor, Leningrad Region, Russia

## ABSTRACT

Topic modeling is a widely used instrument for the analysis of large text collections. In the last few years, neural topic models and models with word embeddings have been proposed to increase the quality of topic solutions. However, these models were not extensively tested in terms of stability and interpretability. Moreover, the question of selecting the number of topics (a model parameter) remains a challenging task. We aim to partially fill this gap by testing four well-known and available to a wide range of users topic models such as the embedded topic model (ETM), Gaussian Softmax distribution model (GSM), Wasserstein autoencoders with Dirichlet prior (W-LDA), and Wasserstein autoencoders with Gaussian Mixture prior (WTM-GMM). We demonstrate that W-LDA, WTM-GMM, and GSM possess poor stability that complicates their application in practice. ETM model with additionally trained embeddings demonstrates high coherence and rather good stability for large datasets, but the question of the number of topics remains unsolved for this model. We also propose a new topic model based on granulated sampling with word embeddings (GLDAW), demonstrating the highest stability and good coherence compared to other considered models. Moreover, the optimal number of topics in a dataset can be determined for this model.

## INTRODUCTION

Topic modeling is widely used in various areas that require big data clustering, especially when analyzing text collections. In practice, however, many topic models generate uninterpretable topics that require users to measure the coherence of the model. Moreover, many topic models possess a certain level of semantic instability, which means that different runs of the algorithm on the same source data lead to different solutions, and this problem is still open. Furthermore, not for every model, it is possible to determine the correct number of topics in a dataset. However, the model parameter 'number of topics', which

Corresponding author
Sergei Koltcov, skoltsov@hse.ru

determines the level of granularity of the cluster solution, has to be set manually in an explicit form. Recent research (*Koltcov et al., 2021*; *Koltcov et al., 2020*) has shown that models supporting an automatic setting of the number of topics perform poorly and depend on other hidden parameters, for example, the concentration parameter. Nevertheless, using domain knowledge, for example, word embeddings or combinations of neural network layers, seems promising in solving the above problems. Currently, several topic models with neural network elements have been proposed. However, systematic research of such models in terms of quality measures, such as interpretability and stability, and the possibility of determining the correct number of topics, was not conducted. Commonly, researchers concentrate only on interpretability ignoring the problems of stability and the problem of the number of topics. The tuning of topic models is usually based on empirical criteria, which have not been tested on labeled datasets. Thus, the main goal of this work is to fill this gap partially and to test four well-known and available to a wide range of users neural topic models such as (1) embedded topic model (ETM) (*Dieng, Ruiz & Blei, 2020*), (2) Gaussian Softmax distribution (GSM) model (*Miao, Grefenstette & Blunsom, 2017*), (3) Wasserstein autoencoders with Dirichlet prior (W-LDA) (*Nan et al., 2019*), and (4) Wasserstein autoencoders with Gaussian Mixture prior (WTM-GMM) (https://zll17.github.io/2020/11/17/Introduction-to-Neural-Topic-Models#WTM-GMM). Moreover, we propose and test a new topic model called a granulated topic model with word embeddings (GLDAW). To test the models, we consider three labeled test datasets and two levels of their pre-processing. Also, we investigate the influence of six different word embeddings (three in Russian and three in English) on ETM and GLDAW models. To estimate the different features of the models, we calculate three measures: coherence, stability, and Renyi entropy.

To simplify the structure of this work, an overview of topic models with elements of neural networks is provided in Appendix A. An overview of different types of word embeddings is presented in Appendix B. Let us note that word embeddings have become an important tool in the field of natural language processing (NLP) allowing translation of text data into numeric vectors that can be further processed by various algorithms.

The rest of our work consists of the following parts. 'Measures in the field of topic modeling' describes measures that we use to estimate models under study. 'Granulated topic model with word embeddings' describes the new proposed granulated model with word embeddings. This model is an extension of the 'granulated topic model' (*Koltcov et al., 2016a*), which additionally incorporates a semantic context contained in word embeddings. 'Computational experiments' contains the description of the datasets and types of used word embeddings and outlines the design of our computer experiments for each of the considered models. 'Results' describes the results of computer experiments for each model in terms of the chosen quality measures. 'Discussion' compares the obtained results for each model. 'Conclusions' summarizes our findings.

## MEASURES IN THE FIELD OF TOPIC MODELING

To estimate considered topic models, we focus on three measures that reflect different model properties: coherence, stability, and Renyi entropy. An extensive overview of

different quality measures in the field of topic modeling can be found in works *Rüdiger et al. (2022)* and *Chauhan & Shah (2021)*. Below, each of the chosen measures is described in more detail.

## Coherence

Coherence allows one to evaluate the consistency of the model, *i.e.,* how often the most probable words of a topic co-occur in the documents. In other words, it evaluates how strongly the words in the topic are related to each other. Thus, coherence reflects the interpretability of the inferred topics. We have used the "u_mass" version of the coherence measure. This measure can be expressed as follows (*Mimno et al., 2011*):

$$C(t, W(t)) = \sum_{m=2}^{M} \sum_{l=1}^{m-1} \log\left(\frac{D(v_m^t, v_l^t) + 1}{D(v_l^t)}\right),$$

$W(t) = (v_1^t, \ldots, v_M^t)$ is a list of $M$ most probable words in topic $t$, $D(v)$ is the number of documents containing word $v$, and $D(v, v')$ is the number of documents where words $v$ and $v'$ co-occur. Thus, coherence is the sum of logs of the ratio of the number of documents with two co-occurring words to the total number of documents with one of the two evaluated words. In other words, if there are highly probabilistic words in a topic, that have a high co-occurrence in highly probabilistic documents, coherence is large, and the topic is well interpretable. In our numerical experiments, we use Gensim library for the calculation of coherence (*Rehurek & Sojka, 2010*).

## Stability

Prior to defining the stability of a topic model, it is necessary to define the similarity of two topics. We use the definition proposed in *Koltcov, Koltsova & Nikolenko (2014)*. The computation of similarity is based on the normalized Kullback–Leibler divergence: $Kn = (1 - \frac{K}{Max}) \cdot 100\%$, where $Max$ is the maximum value of symmetric Kullback–Leibler divergence, $K$ is symmetric Kullback–Leibler divergence, which is expressed as $K(t_1, t_2) = \frac{1}{2}\left(\sum_w \phi_{wt_1} \ln(\phi_{wt_1}) - \sum_w \phi_{wt_1} \ln(\phi_{wt_2})\right) + \frac{1}{2}\left(\sum_w \phi_{wt_2} \ln(\phi_{wt_2}) - \sum_w \phi_{wt_2} \ln(\phi_{wt_1})\right)$ for topics $t_1$ and $t_2$, $\phi_{\cdot t_1}$ is the probability distribution of words in the first topic, $\phi_{\cdot t_2}$ is the probability distribution of words in the second topic. As demonstrated in *Koltcov, Koltsova & Nikolenko (2014)*, two topics can be considered semantically similar if $Kn > 90\%$, meaning that the most probable 50–100 words of these topics are the same. In our work, a topic is considered stable if it is reproduced in three runs of the model with the same number of topics and normalized Kullback–Leibler divergence is not less than 90%. Then, the number of such stable topics is counted. This number depends on the model architecture, the total number of topics, which is a model parameter, and the type of used word embeddings. Therefore, in our experiments, we varied these factors.

## Renyi entropy

The complete derivation of Renyi entropy can be found in *Koltcov (2017)* and *Koltcov (2018)*. In this article, we give the basic idea behind the deformed Renyi entropy for determining the optimal number of topics. The matrix of distribution of words in topics

is one of the results of topic modeling. In practice, researchers work with the most probable words in every topic. These highly probabilistic words can be used to compute the following quantities: (1) Density-of-states function, $\rho = N/(WT)$, where $N$ is the number of words with high probabilities, $W$ is the number of words in the vocabulary, and $T$ is the number of topics. We define "high probability" as a probability larger than $1/W$. (2) Energy of the system $E = -T \cdot \ln\tilde{P} = -T \cdot \ln(\frac{1}{T}\sum_{w,t}(\phi_{wt} \cdot 1_{\{\phi_{wt}>1/W\}}))$, where the summation is over all highly probabilistic words and all topics. Renyi entropy can be expressed as follows: $S_q^R = \frac{q\ln(\tilde{P})+\ln(\rho)}{q-1}$, where the deformation parameter $q = 1/T$ is the inverse number of topics (*Koltcov, 2018*). Since entropy can be expressed in terms of information of the statical system (S = -I (*Beck, 2009*)), a large value of deformed entropy corresponds to a small amount of information and vice versa. Due to the fact that the set of highly probabilistic words in different topic solutions changes with a variation in the total number of topics and other model hyperparameters, *Koltcov, Ignatenko & Koltsova (2019)* and *Koltcov et al. (2020)* discovered the following. First, a small number of topics leads to a very large Renyi entropy, meaning that such a model is poor in terms of information. Second, a significant increase in the number of topics leads to a large entropy as well because topic modeling generates solutions with almost uniform distributions, *i.e.*, topics are indistinguishable. Therefore, the information of such a system is small as well. Experiments have shown that Renyi entropy has a minimum at a certain number of topics, which depends on the particular dataset. Moreover, experiments on labeled datasets have shown that this minimum corresponds to the number of topics obtained with manual labeling. Thus, Renyi entropy can be used to determine the optimal number of topics. It should be noted that this approach is suitable for the datasets with a "flat topic structure" and for the datasets with a hierarchical structure (*Koltcov et al., 2021*).

## GRANULATED TOPIC MODEL WITH WORD EMBEDDINGS

Before describing our proposed topic model, it is necessary to introduce basic notations and assumptions. Let $D$ be a collection of documents, and let $\tilde{W}$ be the set of all words (vocabulary). Each document $d$ is represented as a set of words $w_1, ... w_{n_d}$, $w_i \in \tilde{W}$. The key assumption of probabilistic topic models is that each word $w$ in a document $d$ is associated with some topic $t \in \tilde{T}$, and the set of such topics $\tilde{T}$ is finite. Further, the set of documents is treated as a collection of random independent samples $(w_i, d_i, z_i), i = 1 .. n$, from a discrete distribution $p(w, d, z)$ on the finite probability space $\tilde{W} \times D \times \tilde{T}$. Words and documents are observable variables, and the topic $z \in \tilde{T}$ of every word occurrence is a hidden variable. In topic models, documents are represented as bags of words, disregarding the order of words in a document and the order of documents in the collection. The basic assumption here is that specific words occurring in a document depend only on the corresponding topic occurrences and not on the document itself. Thus, it is supposed that $p(w|d)$ can be represented as $p(w|d) = p(w|t)p(t|d) = \phi_{wt}\theta_{td}$, where $\phi_{wt} = p(w|t)$ is the distribution of words by topics and $\theta_{td} = p(t|d)$ is the distribution of topics by documents. Therefore, to train a topic model on a set of documents means to find the set of topics $\tilde{T}$, and more precisely, to find the distributions $\phi_{wt}, t \in \tilde{T}$ and $\theta_{td}, d \in D$. Let us denote by

matrix $\Phi = \{\phi_{wt}\}$ the set of distributions of words by topics and by matrix $\Theta = \{\theta_{td}\}$ the set of distributions of topics in the documents. There are two major approaches to finding $\Phi$ and $\Theta$. The first approach is based on an algorithm with expectation–maximization inference. The second approach is based on an algorithm that calculates probabilities *via* the Monte-Carlo method. A detailed description of the models and types of inferences can be found in the recent reviews of topic models (*Helan & Sultani, 2023*; *Chauhan & Shah, 2021*).

## Granulated latent Dirichlet allocation model

The granulated topic model is based on the following ideas. First, there is a dependency between a pair of unique words, but unlike the convolved Dirichlet regularizer model (*Newman, Bonilla & Buntine, 2011*), this dependency is not presented as a predefined matrix. Instead, it is assumed that a topic consists of words that are not only described by a Dirichlet distribution but also often occur together; that is, we assume that words that are characteristic for the same topic are often collocated inside some relatively small window (*Koltcov et al., 2016a*). That means all words inside a window belong to one topic or a small set of topics. As previously described in *Koltcov et al. (2016a)*, each document can be treated as a grain surface consisting of granules, which, in turn, are represented as sequences of subsequent words of some fixed length. The idea is that neighboring words usually are associated with the same topic, which means that topics in a document are not distributed independently but rather as grains of words belonging to the same topic.

In general, the Gibbs sampling algorithm for local density of the distribution of words in topics can be formulated as follows:

- Matrices $\Theta$ and $\Phi$ are initialized.
- Loop on the number of iterations

    - For each document $d \in D$ repeat $|d|$ times:

        * sample an anchor word $w_j \in d$ uniformly at random
        * sample its topic $t$ as in Gibbs sampling (*Griffiths & Steyvers, 2004*)
        * set $t_i = t$ for all $i$ such that $|i - j| \leq l$, where $l$ is a predefined window size.

In the last part of the modeling, after the end of sampling, the matrices $\Phi$ and $\Theta$ are computed from the values of the counters. Thus, the local density function of words in topics and the size of the window work as a regularization. The main advantage of this model is that it has very high stability and outperforms other models such as ARTM (*Vorontsov, 2014*), LDA (E-M algorithm) (*Blei, Ng & Jordan, 2003*), LDA with Gibbs sampling algorithm (*Griffiths & Steyvers, 2004*) and pLSA (*Hofmann, 1999*) in terms of stability (*Koltcov et al., 2016b*).

## Granulated latent Dirichlet allocation with word embeddings

A significant disadvantage of the GLDA model is that the Renyi entropy approach for determining the number of topics is not accurate for this model (*Koltcov, 2018*). In this work, we propose a new granulated model (GLDAW), which takes into account information from word embeddings. The GLDAW model is realized with Gibbs sampling algorithm

as follows. There are three stages of the algorithm. In the first stage, we form a matrix of the nearest words by given word embeddings. At this stage, the algorithm checks if the vocabularies of the dataset and word embeddings match. If a word from the given set of word embeddings is missing from the dataset's vocabulary, then its embedding is deleted. This procedure reduces the size of the set of word embeddings and speeds up the computation at the second stage. Then, the matrix of the nearest words in terms of their embedding vectors is built. The number of the nearest words is set manually by the "window" parameter. In fact, this parameter is analogous to the "window" parameter in GLDA.

In the second stage, the computation is similar to Granulated LDA Gibbs sampling with the choice of an anchor word from the text and the attachment of this word to a topic. The topic is computed based on the counters. However, unlike the granulated version, where the counters of the nearest words (to the current anchor word) in the text were increased, this algorithm increases the counters of the words corresponding to the nearest embeddings. Thus, running through all the documents and all the words, we create the matrix of the counters of words taking into account their embeddings.

In the third stage, the resulting matrix of counters is used to compute the probabilities of all words as in the standard LDA Gibbs sampling: $\phi_{wt} = \frac{n_{wt}+\beta}{n_t+\beta W}$, $\theta_{td} = \frac{n_{td}+\alpha}{n_d+\alpha T}$, where $n_{wt}$ equals how many times word $w$ appeared in topic $t$, $n_t$ is the total number of words assigned to topic $t$, $n_{td}$ equals how many times topic $t$ appeared in document $d$. Thus, this procedure of sampling resembles the standard LDA Gibbs sampling, for which the Renyi entropy approach works accurately, but it also has the features of granulated sampling leading to the high stability of the model. It should be noted that the proposed sampling does not have any artificial assumptions about the distribution of topics, as in 'Embedded topic model' (*Dieng, Ruiz & Blei, 2020*), for example, where the topics are sampled from a categorical distribution with parameters equal to the dot product of the word and topic vectors. The proposed sampling procedure uses only information about the closeness of word embeddings.

## COMPUTATIONAL EXPERIMENTS

A short description of the models (ETM, GSM, W-LDA, WTM-GMM, and GLDAW) used in computational experiments is given in Table 1. For a more detailed description of the models, we refer the reader to Appendix A.

To test the above models, the following datasets were used:

- The 'Lenta' dataset is a set of 8,630 news documents in Russian language with 23,297 unique words. The documents are manually marked up into 10 classes. However, some of the topics are close to each other and, therefore, this dataset can be described by 7–10 topics.
- The '20 Newsgroups' dataset is a collection of 15,425 news articles in English language with 50,965 unique words. The documents are marked up into 20 topic groups. According to *Basu, Davidson & Wagstaff (2008)*, 14–20 topics can describe the documents of this dataset, since some of the topics can be merged.

**Table 1  Summary of the models used in numerical experiments.**

| Model | Short description | Word embeddings |
|---|---|---|
| ETM | A log-linear model that takes the inner product of the word embedding matrix ($\rho$) and the topic embedding ($\alpha_k$): proportions of topics $\theta_d \sim LN(0, I)$ ($LN$ means logistic-normal distribution), topics $z_{dn} \sim Cat(\theta_d)$ ($Cat$ means categorical distribution), words $w_{dn} \sim softmax(\rho^T \alpha_{z_{dn}})$. The architecture of the model is a variational autoencoder. | yes |
| GSM | Proportions of topics ($\theta_d$) are set with Gaussian softmax, topics $z_{dn} \sim Multi(\theta_d)$ ($Multi$ refers to the multinomial distribution), words $w_{dn} \sim Multi(\beta_{z_{dn}})$ ($\beta_{z_{dn}}$ is the distribution of words in topic $z_{dn}$.) The architecture of the model is a variational autoencoder. | no |
| W-LDA | The prior distribution of the latent vectors $z$ is set as Dirichlet distribution, while the variational distribution is regulated under the Wasserstein distance. The architecture is a Wasserstein autoencoder. | no |
| WTM-GMM | An improved model of the original W-LDA. The prior distribution is set as Gaussian mixture distribution. The architecture is a Wasserstein autoencoder. | no |
| GLDAW | It is assumed that words in a topic not only follow Dirichlet distribution but also that the words with near embeddings often co-occur together. The inference is based on Gibbs sampling. | yes |

- The 'WoS' dataset is a class-balanced dataset, which contains 11,967 abstracts of published papers available from the *Web of Science*. The vocabulary of the dataset contains 36,488 unique words. This dataset has a hierarchical markup, where the first level contains seven categories, and the second level consists of 33 areas.

The datasets with lemmatized texts used in our experiments are available at https://doi.org/10.5281/zenodo.8407610. For experiments with Lenta dataset we have used the following Russian-language embeddings: (1) Navec are compact Russian embeddings (the part of "Natasha project" https://github.com/natasha/navec), (2) 300_wiki embeddings are fastText embeddings for Russian language(https://fasttext.cc/docs/en/pretrained-vectors.html, *Bojanowski et al., 2017*), (3) Rus_vectors embeddings (RusVectores project) are available at https://rusvectores.org/en/. For experiments with 20 Newsgroups and WoS datasets, we have used the following English-language embeddings: (1) Crawl-300d-2M (fastText technology), available at https://fasttext.cc/docs/en/english-vectors.html (*Mikolov et al., 2018*), (2) Enwiki_20180420_win10_100d are word2vec embeddings(https://pypi.org/project/wikipedia2vec/0.2/, version 0.2) (*Yamada et al., 2016*) (3) wiki_news_300d-1M are fastText embeddings (https://fasttext.cc/docs/en/english-vectors.html (*Mikolov et al., 2018*)).

Numerical experiments were carried out as follows. Every dataset has gone through two levels of pre-processing. At the first level, the words consisting of three or fewer letters were removed. At the second level, additionally, the words that appear less than five times were removed. Thus, the size of Lenta dataset after the first stage of pre-processing is 17,555

unique words and after the second stage of pre-processing is 11,225 unique words. The size of 20 Newsgroups dataset is 41,165 and 40,749 unique words, correspondingly. The size of WoS dataset is 31,725 and 14,526 unique words for two levels of pre-processing. For two types of pre-processing, the experiments have been carried out separately.

For the ETM and GLDAW models, three Russian-language and three English-language word embeddings were used. In the ETM model, pre-trained word embeddings can be trained additionally during the process of modeling. Therefore, this model was tested with and without additional training of word embeddings. The window size for GLDAW model was varied as follows: 10, 50, and 100 for Lenta and 20 Newsgroups datasets; 10, 30, and 50 for WoS dataset.

The number of topics was varied in the range [2; 50] in increments of one topic for all models. All three chosen measures were calculated as a mean of three runs of each model. Source codes of our computational experiments are available at https://doi.org/10.5281/zenodo.8410811.

## RESULTS

Numerical results are described according to the chosen measures. So, in the first part, we describe the results of all five models in terms of coherence. The second part analyzes the stability of the models. In the third part, we assess the possibility of determining the true number of topics. Finally, in the fourth part, we present the computational speed of each model on an example of WoS dataset.

### Results on coherence

As discussed, the ETM model was trained according to two schemes: (1) with pre-trained word embeddings and (2) with additionally trained embeddings. Figure 1 demonstrates the results of ETM model with different types of word embeddings in terms of coherence measure. Let us note that for some numbers of topics, there is no coherence value for pre-trained embeddings on the first level of pre-processing for WoS dataset, which means that the model performs poorly with these settings.

Based on our results (Fig. 1), one can conclude the following. First, training embeddings during the model learning leads to better coherence for all three datasets. Moreover, in this case, the coherence values are almost identical for different types of embeddings for each dataset. Second, ETM model performs poorly on a small dataset (11–12 thousand words) with strong pre-processing (Fig. 1B). Therefore, larger datasets should be used for ETM model to get better quality. Third, the behavior of the coherence measure does not allow us to estimate the optimal number of topics for this model.

Let us note that the GSM, W-LDA and WTM-GMM models do not use word embeddings. The results for these three topic models are given in Fig. 2. Overall, our results demonstrate that the dataset pre-processing does not strongly influence the output of the above models. The GSM model performs worse than the other models, while WTM-GMM shows the best results. However, the fluctuation of coherence is larger for all these models than for ETM model.

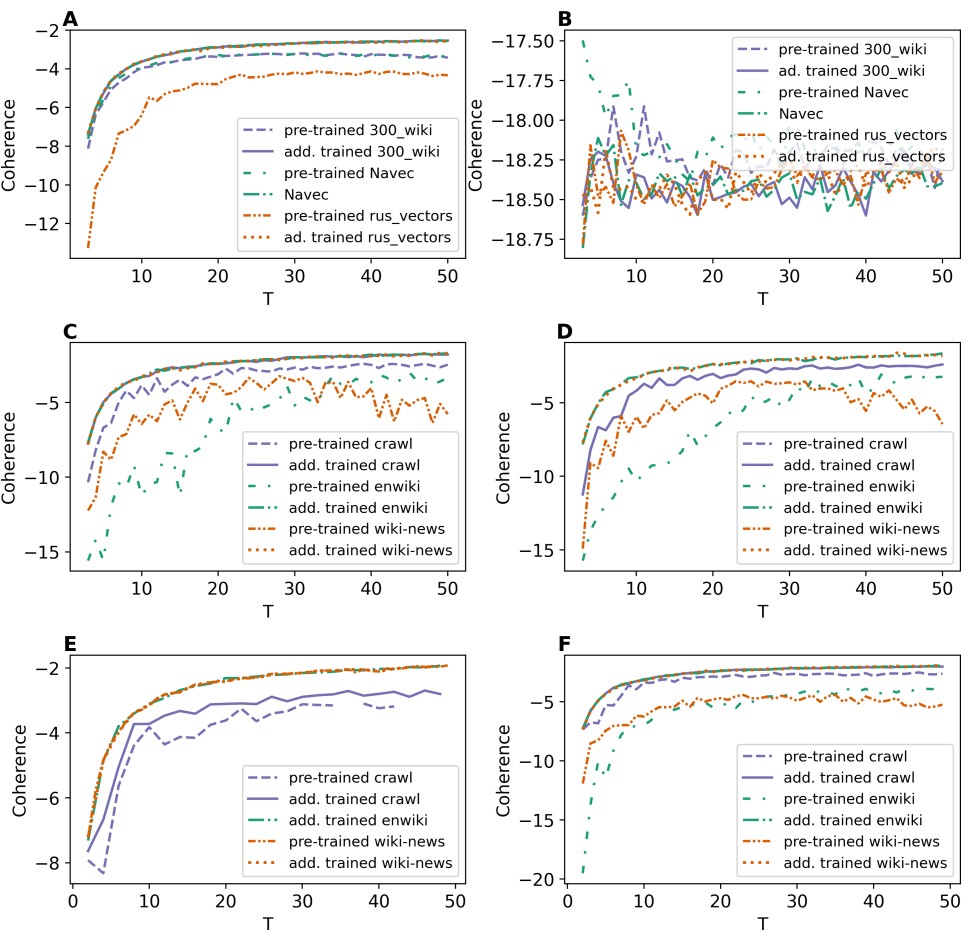

**Figure 1** **Dependence of coherence on the number of topics (ETM model).** (A) Lenta dataset: first level of pre-proccessing. (B) Lenta dataset: second level of pre-proccessing. (C) 20 Newsgroups dataset: first level of pre-processing. (D) 20 Newsgroups dataset: second level of pre-processing. (E) WoS dataset: first level of pre-processing. (F) WoS dataset: second level of pre-processing.

Figure 3 demonstrates the results of GLDAW model. Figures 3A and 3B show that pre-processing of Lenta dataset does not have a strong influence on the performance of this model. Besides that, the coherence values do not depend on the type of embeddings. The fluctuation for different embeddings is about 0.1, while for W-LDA, WTM-GMM, and GSM models, the fluctuation is about 1. The fluctuation of the coherence value of ETM model for trained embeddings is also about 0.1. The values of coherence for the 20 Newsgroups dataset are given in Figs. 3C and 3D. The fluctuation of coherence values for all types of embeddings is about 0.4 on the first level of pre-processing and about 0.3 on the second level of pre-processing. For the WoS dataset (Figs. 3E, 3F), the fluctuation of coherence for all types of embeddings is about 0.3 for $T < 25$ and about 0.5 for $T > 25$ on the first level of pre-processing. On the second level of pre-processing, the fluctuation of coherence does not exceed 0.3. Thus, the GLDAW model has a smaller fluctuation of coherence in comparison to considered neural topic models. Also, the quality almost does

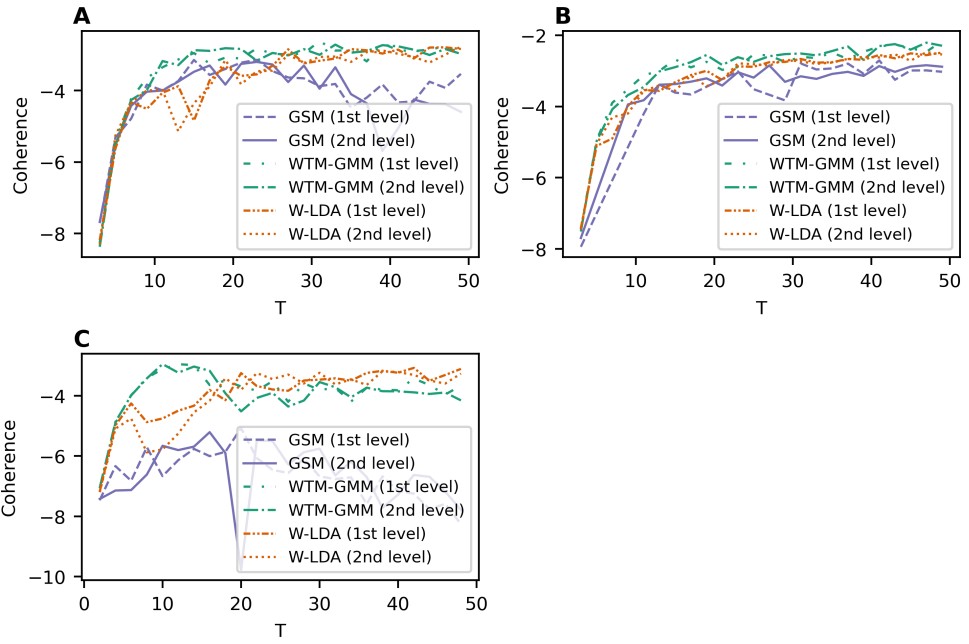

**Figure 2** Dependence of coherence on the number of topics (W-LDA, WTM-GMM, and GSM models). (A) Lenta dataset. (B) 20 Newsgroups dataset. (C) WoS dataset.

not depend on the size of the window. The ETM model has a similar fluctuation value, but it does not perform well for the deeply pre-processed small datasets, while the GLDAW performance is good for both levels of pre-processing.

To compare all five models, Fig. 4A demonstrates the best coherence values for Lenta dataset. ETM model shows the best quality with Navec embeddings at the first level of pre-processing. However, this model does not perform well on the second level of pre-processing (Fig. 1C). The GLDAW model has the second-best result with values of 0.1 less than ETM but performs well on the second level of pre-processing. Besides that, the GLDAW model does not require the additional training of embeddings, unlike the ETM model.

The best models in terms of coherence for 20 Newsgroups dataset are presented in Fig. 4B. The best result is achieved by ETM as well. The GLDAW model outperforms W-LDA in the range of 2–20 topics, while W-LDA outperforms GLDAW in the range of 30–50 topics. It should be noted that the optimal number of topics is 14–20 for this dataset, according to human judgment. Figure 4C demonstrates the best results for the WoS dataset. GLDAW outperforms other models in terms of coherence, while the ETM model has the second-best results for this dataset.

## Results on stability

Let us remind that stability was computed based on the three runs of every model with the same settings. The topic is considered stable if it is reproduced in all runs with a normalized Kullback–Leibler divergence above 90%. Otherwise, the topic is considered unstable. The number of stable topics was calculated for topic solutions with a fixed total number of

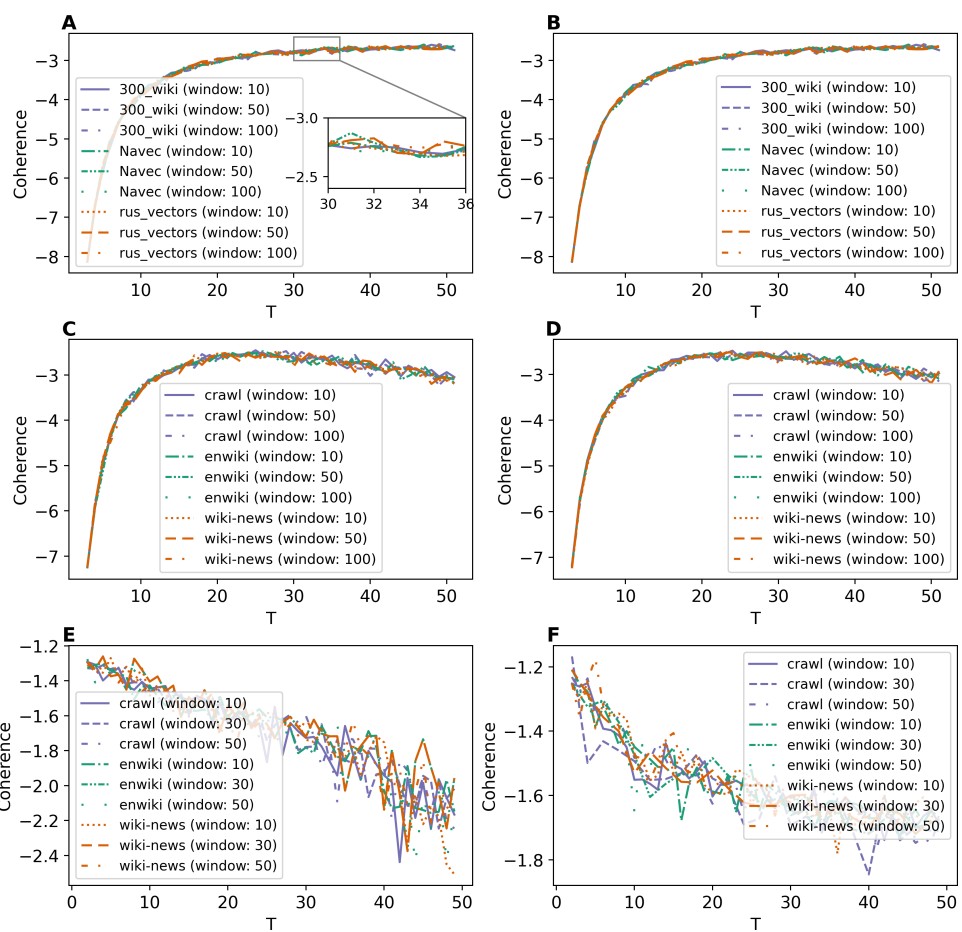

**Figure 3** **Dependence of coherence on the number of topics (GLDAW model).** (A) Lenta dataset: first level of pre-proccessing. (B) Lenta dataset: second level of pre-proccessing. (C) 20 Newsgroups dataset: first level of pre-processing. (D) 20 Newsgroups dataset: second level of pre-processing. (E) WoS dataset: first level of pre-processing. (F) WoS dataset: second level of pre-processing.

topics. Figures 5–7 demonstrate the number of stable topics *vs.* the total number of topics in topic solutions. Since the calculation of stability is a very time-consuming procedure, we considered only $T = 10, 20, 30$ for WoS dataset.

Figures 5A and 5B demonstrate the results on the number of stable topics for Lenta dataset. One can see that additional training of word embeddings significantly improves the model stability for a large number of topics. However, for a small number of topics, the difference is insignificant. On average, "rus_vectors" embeddings demonstrate the best result. Let us note that the stability measure based on Kullback–Leibler divergence does not allow us to determine the optimal number of topics for Lenta dataset.

Figures 5C and 5D demonstrate the results on stability for the 20 Newsgroups dataset. One can see that the application of pre-trained embeddings leads to less stable models as well as for the Lenta dataset. For example, the models with wiki-news-300d-1 M and enwiki_20180420 embeddings are stable only in the range of 10–20 topics on the first level

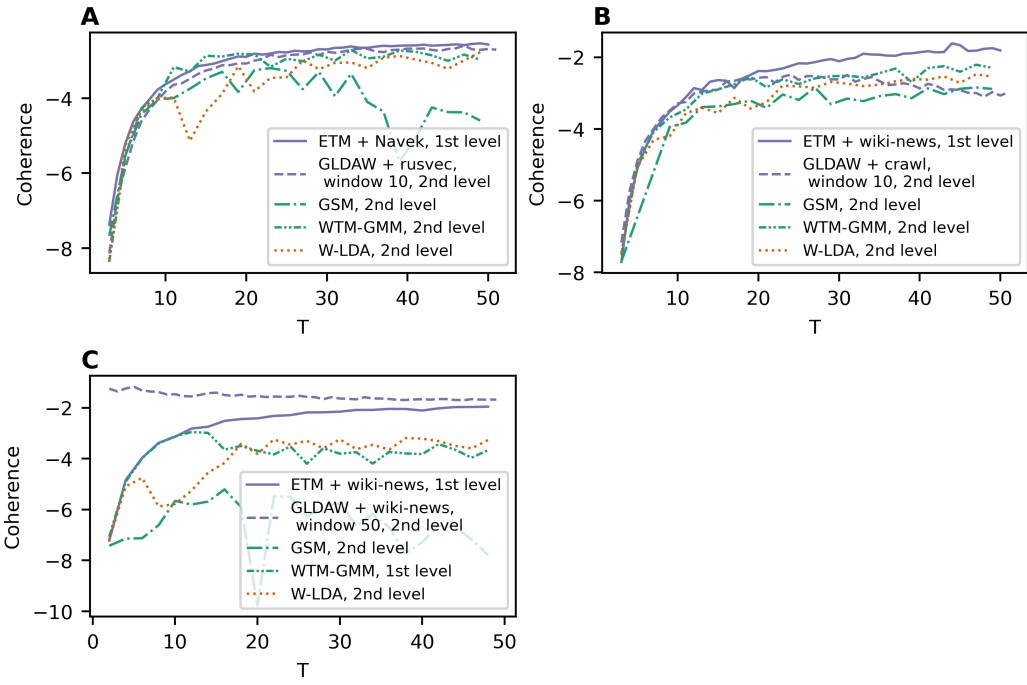

**Figure 4** **Dependence of the best coherence values on the number of topics.** (A) Lenta dataset. (B) 20 Newsgroups dataset. (C) WoS dataset.

of pre-processing. The second level of pre-processing increases stability, and the fluctuation of stability reduces. Also, it should be noted that these curves do not allow us to evaluate the optimal number of topics for 20 Newsgroups dataset. Altogether, we obtain the following results for the above two datasets: for Lenta dataset, ETM model is stable in the range of 1–4 topics, while this dataset has 7–10 topics according to human judgment; for 20 Newsgroups dataset, ETM model is stable in the range of 8–15 topics while this dataset has 14–20 topics according to human judgment.

Figures 5E and 5F demonstrate the results on stability for the WoS dataset. Again, one can see that additional training of embeddings increases the stability of the model. Moreover, one can see that the ETM model is sensitive to the type of word embeddings and may produce a relatively large number of stable topics that are not present in the dataset according to human judgment.

Figure 6 demonstrates the results on stability for the W-LDA, WTM-GMM and GSM models. The level of pre-processing does not affect the stability of these models. However, the stability level of the GSM, W-LDA, and WTM-GMM models is 3–4 times less than the stability of ETM. The GSM model demonstrates the worst results in terms of stability for the 20 Newsgroups dataset. It starts to produce stable solutions only beginning with 10 topics, and in general, it has 3–6 times worse stability than the WTM-GMM and W-LDA models. Let us note that for the WoS dataset, we obtained zero stable topics for most cases. On the whole, all models are stable in the range of 1–3 topics for Lenta dataset and in

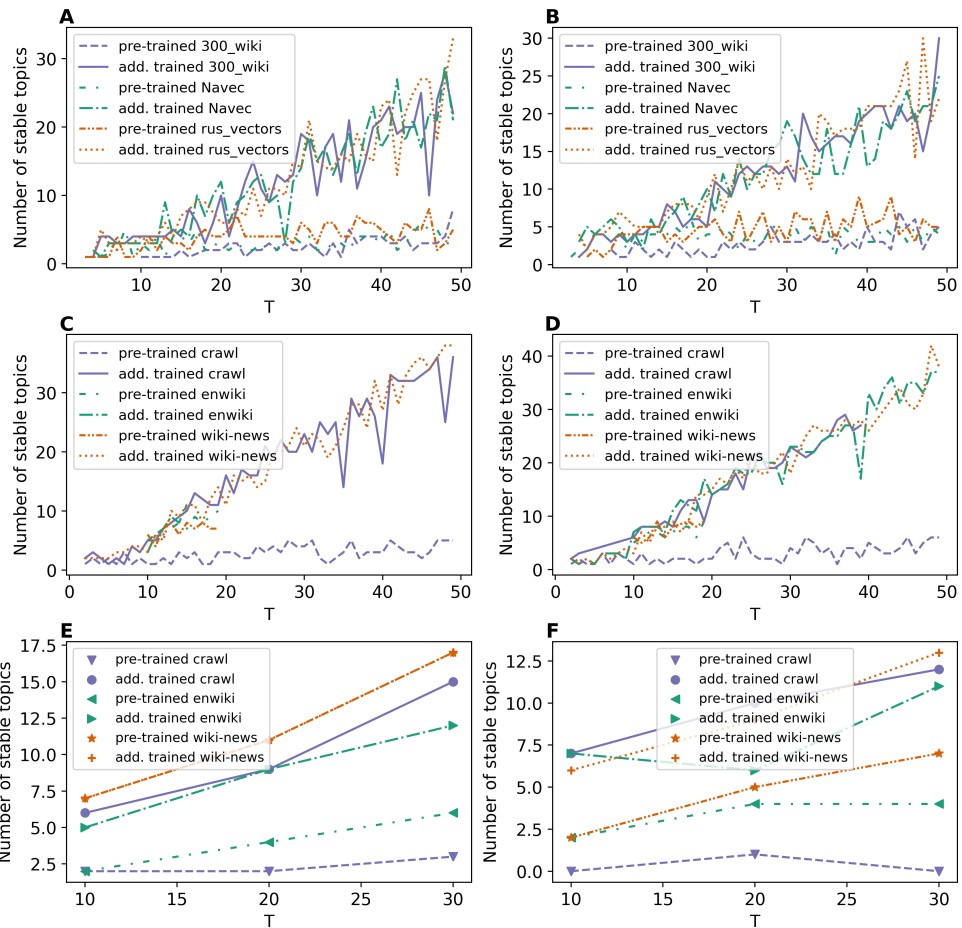

**Figure 5** **Dependence of the number of stable topics on the total number of topics (ETM model).**
(A) Lenta dataset: first level of pre-proccessing. (B) Lenta dataset: second level of pre-proccessing.
(C) 20 Newsgroups dataset: first level of pre-processing. (D) 20 Newsgroups dataset: second level
of pre-processing. (E) WoS dataset: first level of pre-proccessing. (F) WoS dataset: second level of
pre-proccessing.

the range of 9–15 topics for the 20 Newsgroups dataset. However, for the WoS dataset we
obtained the worst results without stable topics for the GSM and W-LDA models.

Figures 7A and 7B demonstrate results on stability for GLDAW model on Lenta dataset.
The levels of pre-processing almost do not influence the results. The number of stable
topics is about 6–8 topics for solutions on 7–10 topics for both levels of pre-processing.
The further increase in the number of topics leads to a small increase in the number of
stable topics up to 16 topics. Figures 7C and 7D show the results for the 20 Newsgroups
dataset. These curves show that the levels of pre-processing do not influence the stability
either. Moreover, the fluctuation of stability is significantly smaller in the region of the
optimal number of topics than in the region of a large number of topics. Furthermore, the
largest number of stable topics that the GLDAW model produces is about 27 topics when
increasing the number of topics. Thus, the GLDAW model does not prone to produce
redundant topics, while the ETM model with 50 topics produces 35 stable topics meaning

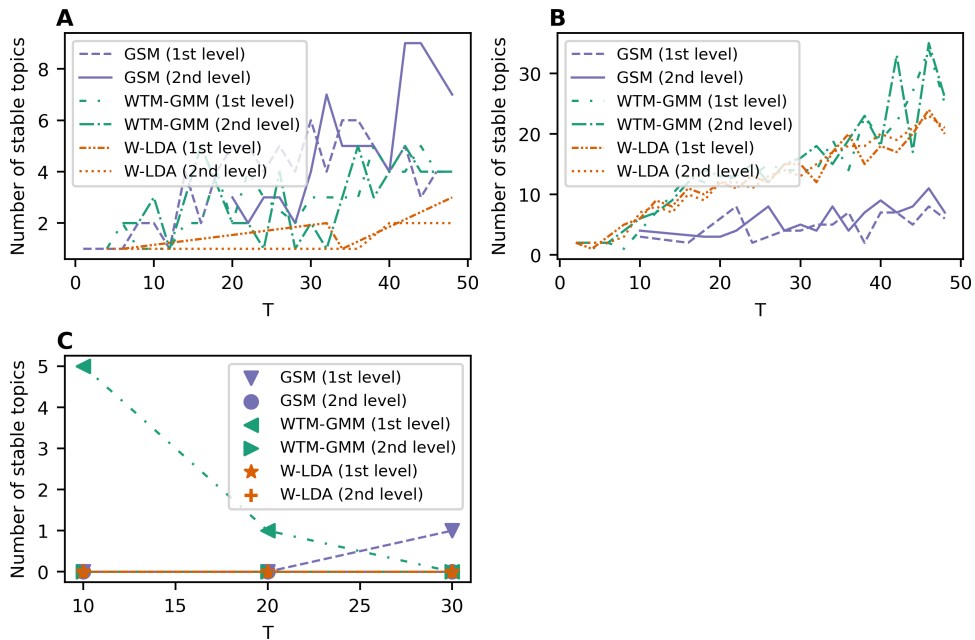

**Figure 6  Dependence of the number of stable topics on the total number of topics (W-LDA, WTM-GMM and GSM models).** (A) Lenta dataset. (B) 20 Newsgroups dataset. (C) WoS dataset.

that it finds redundant topics (since the real number of topics is 14–20). At the same time, the GLDAW model produces 14–18 stable topics in this range. Figures 7E and 7F demonstrate the results on stability for WoS dataset. The number of stable topics varies in the range of 4–11 topics, which is close to the number of topics on the first level of markup. On the whole, the GLDAW model demonstrates the best result in terms of stability among all considered models and for all three datasets. Since this model does not depend significantly on the type of embeddings and is not sensitive to the window size, we recommend using a window size equal to ten in order to speed up the calculation time.

## Results on Renyi entropy

Various studies of topic models (*Koltcov, 2018*; *Koltcov et al., 2020*; *Koltcov et al., 2021*) have shown that the number of topics corresponding to the minimal Renyi entropy equals the optimal number of topics, *i.e.,* the number of topics according to human judgment. There can be several minimal points in the case of hierarchical structure (*Koltcov et al., 2021*). However, in this work, we focus on labeled datasets with a flat structure leading to a single Renyi entropy minimum, and for hierarchical WoS dataset, we consider only the markup on the first level containing categories.

Figures 8A and 8B demonstrate Renyi entropy curves for ETM model on Lenta dataset. First, the figures show that applying pre-trained embeddings results in a minimum occurring at a small number of topics, about 4–6 topics, that does not match the human labeling. Second, additionally trained embeddings shift the minimum in the range of 9–10 topics. Besides that, the difference between models with different embeddings is

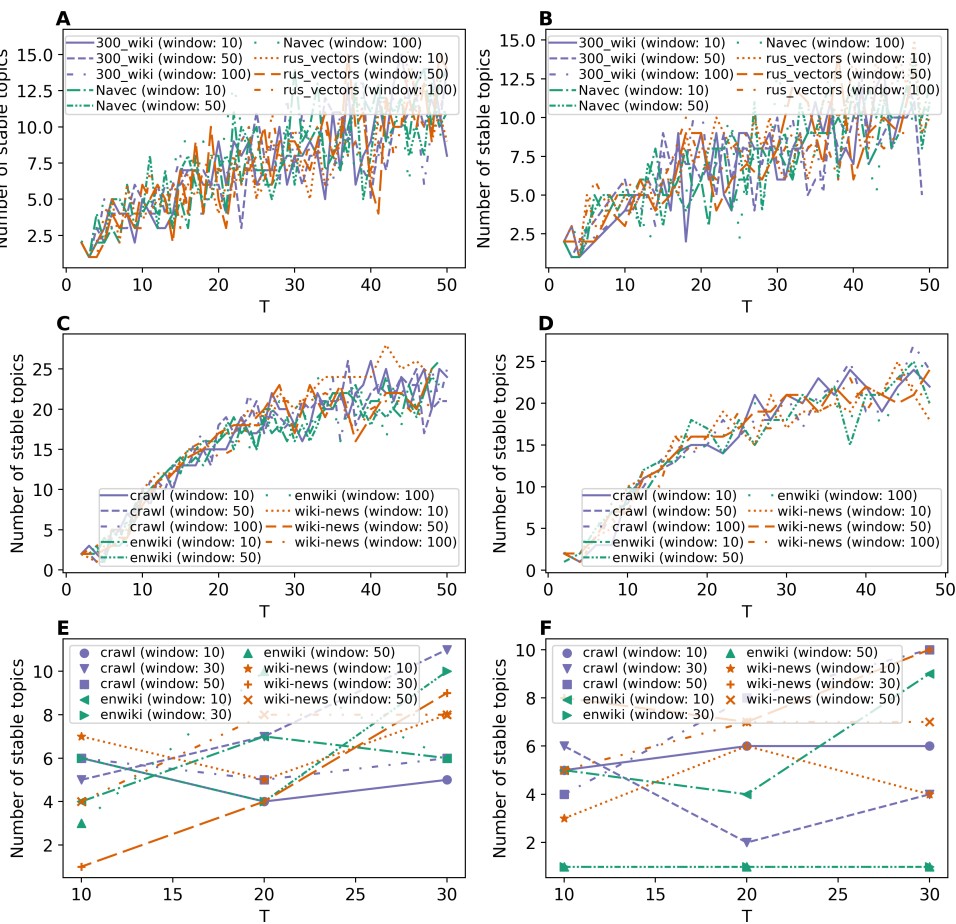

**Figure 7** **Dependence of the number of stable topics on the total number of topics (GLDAW model).**
(A) Lenta dataset: first level of pre-proccessing. (B) Lenta dataset: second level of pre-proccessing.
(C) 20 Newsgroups dataset: first level of pre-processing. (D) 20 Newsgroups dataset: second level
of pre-processing. (E) WoS dataset: first level of pre-proccessing. (F) WoS dataset: second level of
pre-proccessing.

insignificant. Third, the pre-processing almost does not influence the curves for models
with additionally trained embeddings.

Renyi entropy curves for ETM model on the 20 Newsgroups dataset are given in Figs. 8C
and 8D. These figures show that it is necessary to train embeddings. There is no significant
difference between models with different embeddings. The minimum of Renyi entropy is
at 11 topics, which does not match the human mark-up.

Figures 8E and 8F demonstrate Renyi entropy curves for the ETM model on the WoS
dataset. The curves corresponding to pre-trained embeddings have minimum points not
matching the mark-up (2–3 topics for the first level of pre-processing and 3–6 topics for the
second level of pre-processing). The minimum of Renyi entropy for additionally trained
embeddings corresponds to 8–12 topics. Let us note that this number is the same for both
pre-processing levels and all three datasets, meaning that sampling from the categorical
distribution parameterized by the dot product of the word and topic embeddings works as

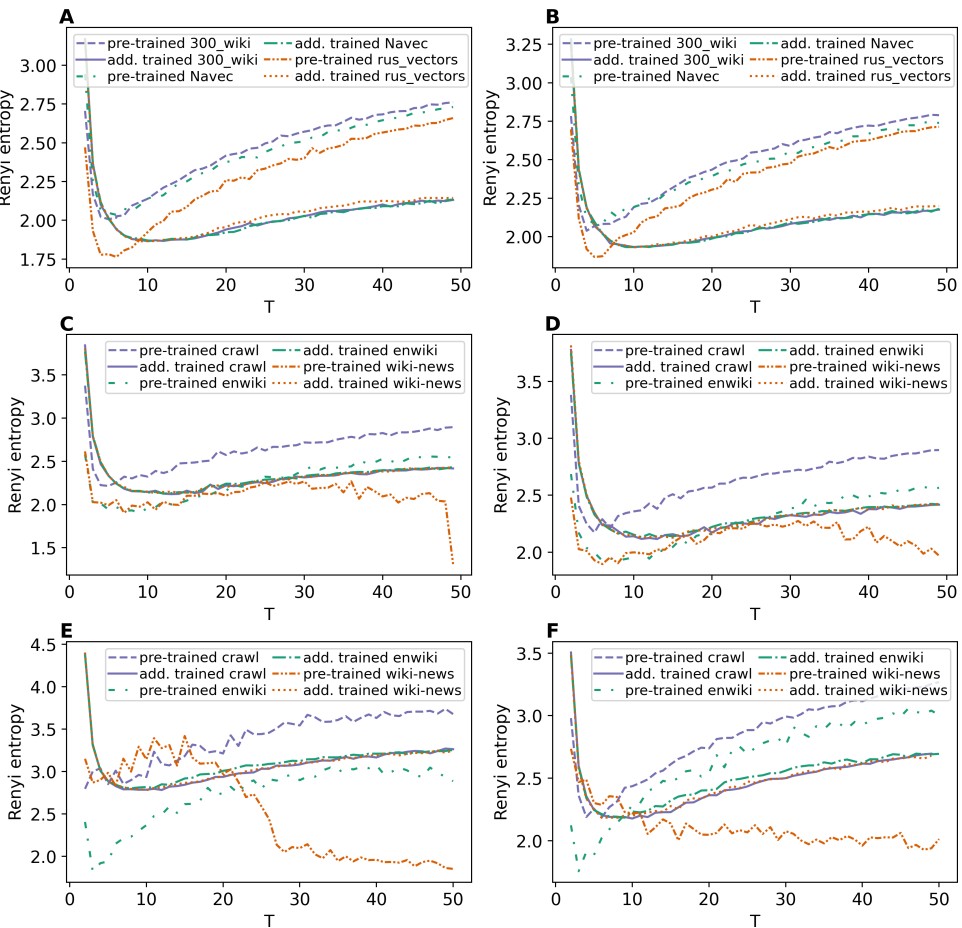

**Figure 8 Dependence of Renyi entropy on the number of topics (ETM model).** (A) Lenta dataset: first level of pre-proccessing. (B) Lenta dataset: second level of pre-processing. (C) 20 Newsgroups dataset: first level of pre-processing. (D) 20 Newsgroups dataset: second level of pre-processing. (E) WoS dataset: first level of pre-processing. (F) WoS dataset: second level of pre-processing.

a too strong regularization. It leads to the fact that different datasets' results do not differ much. Therefore, Renyi entropy cannot be used to determine the optimal number of topics for ETM model.

Renyi entropy curves for the neural topic models trained on Lenta dataset are given in Fig. 9A. The results show that Renyi entropy for W-LDA and GSM models does not depend on the level of pre-processing. The entropy minima for GSM and W-LDA models do not correspond to the optimal number of topics. However, Renyi entropy minimum (six topics) for the WTM-GMM model on the second level of pre-processing almost corresponds to the true number of topics. The results on Renyi entropy for the 20 Newsgroups dataset are given in Fig. 9B. This figure shows that the levels of pre-processing do not influence the entropy values. The minimum for GSM model is at eight topics; for W-LDA is at 12 topics; and for WTM-GMM is at 42 topics. Figure 9C demonstrates corresponding results for the WoS dataset. One can see that again, the pre-processing levels do not significantly influence

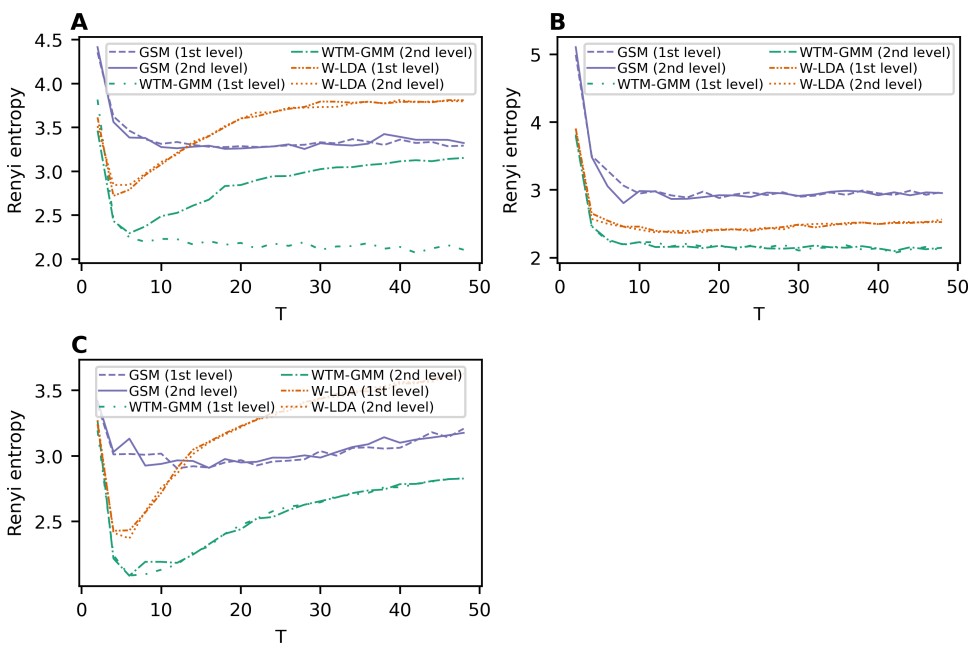

**Figure 9  Dependence of Renyi entropy on the number of topics (W-LDA, WTM-GMM and GSM models).** (A) Lenta dataset. (B) 20 Newsgroups dataset. (C) WoS dataset.

the entropy values. The minimum for the GSM model is at 12-16 topics; for W-LDA is at 4-6 topics; and for WTM-GMM is at six topics. Thus, Renyi entropy minimum for WTM-GMM model is very close to the true number of topics. However, for GSM and W-LDA, the entropy minima do not correspond to the optimal number of topics. Thus, we can conclude that Renyi entropy cannot be used to determine the optimal number of topics for these models.

The Renyi entropy curves for the GLDAW model trained on Lenta dataset are presented in Figs. 10A and 10B. The computations show that the GLDAW model has Renyi entropy minimum in the range of 7–9 topics, and different types of embeddings do not change positions of the minimum. This result matches the number of topics according to the human mark-up. Figures 10C and 10D demonstrate Renyi entropy curves for the 20 Newsgroups dataset. The minimum of Renyi entropy is located in the range of 15-17 topics, which also matches human judgment. Figures 10E and 10F demonstrate Renyi entropy curves for the WoS dataset. The Renyi entropy minimum is achieved in the range of 11–13 topics, and different types of embeddings do not change the position of the minimum. This result is close to the number of topics achieved according to the human mark-up.

Thus, we can conclude that, first, the GLDAW model almost does not depend on the embedding type and window size for both Russian-language and English-language embeddings. Second, this model allows us to correctly determine the approximation of the optimal number of topics for all three datasets.

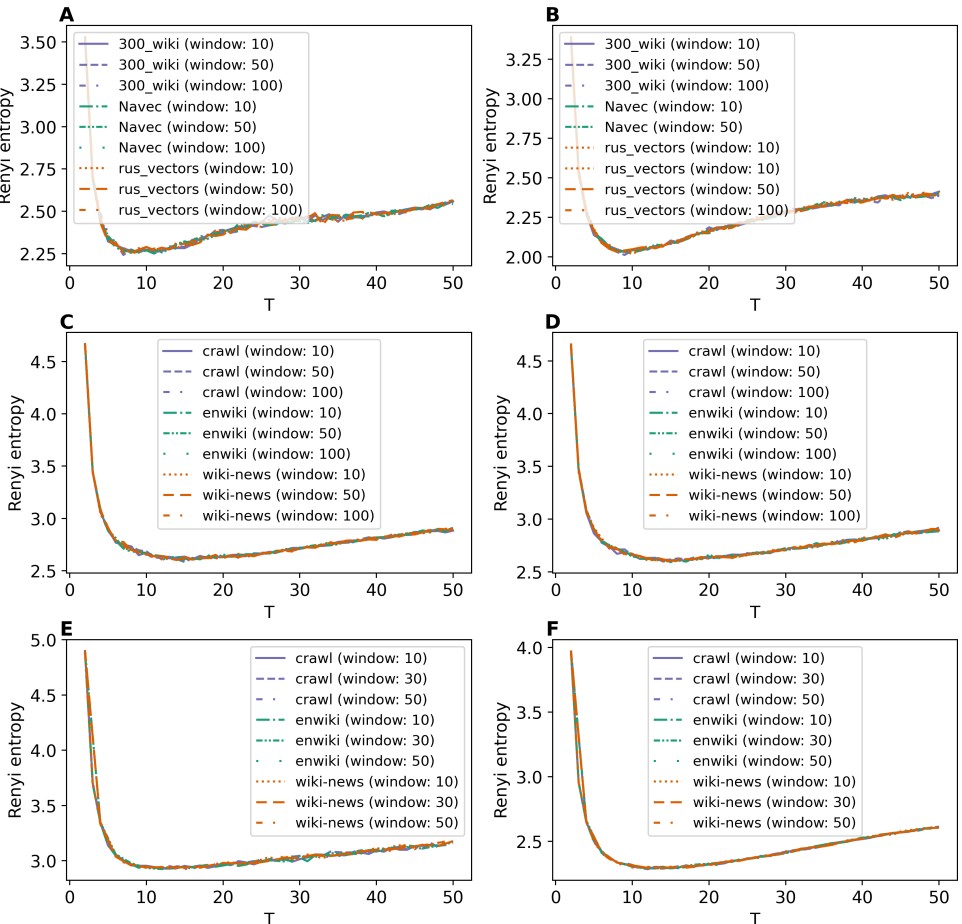

**Figure 10** **Dependence of Renyi entropy on the number of topics (GLDAW model).** (A) Lenta dataset: first level of pre-proccessing. (B) Lenta dataset: second level of pre-proccessing. (C) 20 Newsgroups dataset: first level of pre-processing. (D) 20 Newsgroups dataset: second level of pre-processing. (E) WoS dataset: first level of pre-processing. (F) WoS dataset: second level of pre-processing.

## Computational speed

Table 2 demonstrates the computational speed of the considered models on an example of WoS dataset with the second level of pre-processing for different values of the total number of topics, namely, for $T = 10$, $20$, $30$. The number of epochs for W-LDA, WTM-GMM, GSM, and ETM was fixed at 300, and the number of iterations for GLDAW model was also fixed at 300. All calculations were performed on the following equipment: computer with Intel Core i7-12700H 2.7 GHz, Ram 16 Gb, operation system: Windows 10 (64 bits), graphics card: NVIDIA GeForce RTX 3060. Let us note that the W-LDA, WTM-GMM, GSM, and ETM models were computed with the application of CUDA while the GLDAW model was not optimized for parallel computing.

Based on the presented time costs, one can see that the proposed GLDAW model is the fastest model among the considered ones and, thus, can be recommended for practical use.

**Table 2   Computational speed of the models for WoS dataset.**

| Model | Number of topics | Calculation time |
| --- | --- | --- |
| ETM with pre-trained embeddings (enwiki) | 10 | 248.25 sec |
| ETM with pre-trained embeddings (enwiki) | 20 | 257.62 sec |
| ETM with pre-trained embeddings (enwiki) | 30 | 260.32 sec |
| ETM with additionally trained embeddings (enwiki) | 10 | 250.60 s |
| ETM with additionally trained embeddings (enwiki) | 20 | 261.80 sec |
| ETM with additionally trained embeddings (enwiki) | 30 | 266.94 sec |
| GSM | 10 | 5422.5 sec |
| GSM | 20 | 6696.0 sec |
| GSM | 30 | 6717.0 sec |
| W-LDA | 10 | 6640.0 sec |
| W-LDA | 20 | 6760.6 sec |
| W-LDA | 30 | 6905.9 sec |
| WTM-GMM | 10 | 8299.7 sec |
| WTM-GMM | 20 | 9340.5 sec |
| WTM-GMM | 30 | 9340.8 sec |
| GLDAW with enwiki embeddings, window size 50 | 10 | **41.49 sec** |
| GLDAW with enwiki embeddings, window size 50 | 20 | **47.23 sec** |
| GLDAW with enwiki embeddings, window size 50 | 30 | **53.44 sec** |

# DISCUSSION

## Comparison of models in terms of coherence

Our calculations demonstrated that ETM model has both positive and negative properties in terms of coherence. First, it is necessary to train embeddings for both Russian-language and English-language embeddings to obtain good quality for this model. The procedure of additional training of embeddings is time-consuming, but, in this case, the coherence value is the largest among all considered models for the Lenta and 20 Newsgroups datasets. Second, ETM model performs poorly on datasets with 11,000 words or less. This means that this model is not suitable for small datasets. Moreover, additional training of embeddings does not improve the coherence of the topic model for small datasets.

The GLDAW model has slightly worse results than the ETM model in terms of coherence for the Lenta and 20 Newsgroups datasets; however, it demonstrates the best result for WoS dataset. In addition, this model does not require additional training of embeddings and does not depend on the window size. Moreover, the GLDAW model performs well on small and large datasets, meaning that this model can be used in a wide range of tasks.

The GSM, WTM-GMM, and W-LDA models perform worse than ETM with trained embeddings and GLDAW in terms of coherence. Moreover, these models have significant fluctuations of the coherence measure under variation of the number of topics. The GSM model demonstrates the worst results in terms of coherence for all three datasets.

## Stability of topic models

The best result of the ETM model with trained embeddings in terms of stability for Lenta dataset is about 4–5 topics for topic solutions on 10 topics. For topic solutions on 50 topics,

this model has about 27–30 stable topics. The ETM model with untrained embeddings has only 5–7 stable topics for solutions on 50 topics. Therefore, additional training of word embeddings is required for this model. For the 20 Newsgroups dataset, the results on stability are as follows. The ETM model with untrained embedding demonstrates a limited level of stability for some types of embeddings. For example, for wiki-news-300d-1M and enwiki_20180420 embeddings, the model has stable topics only in the region of 10–20 topics; in other cases, most topics were not reproduced in all three runs. The models with trained embeddings perform better in terms of stability. There are 27–37 stable topics for solutions on 50 topics and 12–15 stable topics (depending on the type of embeddings) for solutions on 20 topics. For the WoS dataset, we can also observe that ETM model with additionally trained embeddings produces a larger number of stable topics, namely, 5–7 stable topics for $T = 10$, 6–11 stable topics for $T = 20$, and 12–17 stable topics for $T = 30$ for both levels of pre-processing.

For the Lenta dataset, the GLDAW model has 7–8 stable topics for solutions on 10 topics outperforming the ETM model. Moreover, it has 8–16 stable topics for solutions on 50 topics, meaning that it is less prone to see redundant topics in this dataset than ETM model, which produces about 30 stable topics. For the 20 Newsgroups dataset, GLDAW model demonstrates the best result in the range of 15–19 topics, which is slightly better than the ETM model with trained embeddings. For topic solutions on 50 topics, GLDAW model has 22–25 stable topics. For WoS dataset, the GLDAW model has 4–11 stable topics for the considered values of $T$, which is close to the human markup. Thus, the GLDAW model outperforms all other models in terms of stability, showing the best stability for the number of topics close to the optimal one, while for the ETM model, the number of stable topics increases as the number of topics increases; what does not match the human mark-up.

The stability of GSM, W-LDA, and WTM-GMM is worse than that of ETM and GLDAW. WTM-GMM shows the best results (among the neural topic models), producing three stable topics for solutions on 10 topics for the Lenta dataset, 13 stable topics for solutions on 20 topics for the 20 Newsgroups dataset, and five stable topics for solutions on 10 topics for the WoS dataset. GSM model demonstrates the worst result in terms of stability.

## Determining the optimal number of topics

ETM model with trained embeddings has a minimum Renyi entropy at 10–11 topics for the Russian-language dataset as well as for the English-language datasets. Thus, the Renyi entropy approach does not allow us to differentiate between datasets for this model and, therefore, to find the optimal number of topics. It happens due to the procedure of sampling. The sampling is made from a categorical distribution parametrized by embeddings, meaning that embeddings have a greater impact on topic solutions than the dataset itself.

The GLDAW model has a minimum Renyi entropy at 7–9 topics for the Lenta dataset, 15–17 topics for the 20 Newsgroups dataset, and 11–13 topics for the WoS dataset, which is close to the human labeling results. Besides that, our computations show that the type of embeddings does not influence the entropy curve behavior for this model. Finally, the

GSM, WTM-GMM, and W-LDA models have a minimum entropy for 4–6 topics for the Russian-language dataset, and a fluctuating minimum at 14–42 topics for 20 Newsgroups dataset. For the WoS dataset, the WTM-GMM model has a minimum for 6 topics, GSM model demonstrates a minimum for 12–16 topics, and W-LDA has a minimum for 4–6 topics.

Thus, according to our results, we can conclude the following. The GLDAW model is the best among all considered models based on the combination of three measures. It has a slightly smaller coherence value than the ETM model for two of three datasets but the largest stability in the region of the optimal number of topics. Moreover, this granulated model allows us to determine the optimal number of topics more accurately than the other models. In addition, this model is the fastest in terms of computational cost among the considered ones.

## CONCLUSIONS

In this work, we have investigated five topic models (ETM, GLDAW, GSM, WTM-GMM, and W-LDA) with elements of neural networks. One of these models, namely the GLDAW model, is new and is based on the granulated procedure of sampling, where a set of nearest words is found according to word embeddings. For the first time, all models were evaluated simultaneously in terms of three measures: coherence, stability, and Renyi entropy. We used three datasets with two levels of pre-processing as benchmarks: the Russian-language dataset 'Lenta', the English-language '20 Newsgroups' and 'WoS' datasets. The experiments demonstrate that the ETM model is the best model in terms of coherence for two of three datasets, while the GLDAW model takes second place for those two datasets and first place for the third dataset. At the same time, the GLDAW model has higher stability than the ETM model. Besides that, it is possible to determine the optimal number of topics in datasets for the GLDAW model, while the ETM model is unable for that. In addition, the GLDAW model demonstrates the smallest computational cost among the considered models. The GSM, WTM-GMM, and W-LDA models demonstrate worse results than the ETM and GLDAW models in terms of all three measures. Thus, the proposed GLDAW model outperforms ETM, WTM-GMM, GSM, and W-LDA in terms of the combination of three quality measures and in terms of computational cost.

## ACKNOWLEDGEMENTS

This research was supported in part through computational resources of HPC facilities at NRU HSE.

## APPENDIX A

A lot of topic models with elements of neural networks, which include network architecture directly as well as application of word embeddings, have been proposed over the past 7 years. In this section, we briefly describe some of these models.

In the Gaussian LDA model (*Das, Zaheer & Dyer, 2015*), documents are represented as sequences of word embeddings obtained with word2vec, while topics are described as multivariate normal distributions in embedding space (in contrast to classical topic model representation, where topics are discrete distributions over the dictionary). Correspondingly, every such distribution is characterized by its mean and variance. The distribution of topics by documents is set as a Dirichlet distribution analogous to the LDA model. Modeling word embeddings, not the words itself, is the distinctive peculiarity of this approach. Different types of word embeddings can be used with this model. Model inference is based on Gibbs sampling. It should be noted that there are a lot of hyperparameters, including the number of topics, which require careful tuning. Also, only the Euclidean distance is used as a proximity measure of embeddings, while the standard approach is to use the cosine measure. This weakness is discussed in *Li et al. (2016c)*. The algorithm implementation of Gaussian LDA model can be found at https://github.com/rajarshd/Gaussian_LDA.

*Nguyen et al. (2015)* incorporate word embeddings into two topic models: LDA and one-topic-per-document DMM model (analogue to LDA, where every document has only one topic) and obtain two new models LF-LDA and LF-DMM, correspondingly. The main purpose of their work is to improve the quality of topic modeling for short texts and small corpuses. The authors use two variants of pre-trained word embeddings: 'GloVe and word2vec. Inference is based on Gibbs sampling. In the framework of htis model, each topic has its representation in the embedding space. The generation of words is as follows. The so-called "unfair coin" (with lambda parameter being responsible for success probability) is tossed for every word. Then, word is generated from standard LDA topic in case of failure. Otherwise, the word embedding generated from the component of the topic vector is taken instead of the word itself. Thus, it is a mixed model, where the word is generated either from the component of the topic, which corresponds to the Dirichlet LDA topic, or from the latent feature component represented as a vector. The representations of topic vectors as well as matrices $\Phi$ and $\Theta$ (similar to LDA) are estimated during the model inference. It is shown that the best PMI-score is given by models with lambda equal to 1 meaning that topics are represented only by their embeddings. The peculiarity of this model is that every topic is a combination of the Dirichlet topic (inferred with LDA) and a vector representation, which can have different top words, however the authors don't mention how to handle it. The model implementation can be found at https://github.com/datquocnguyen/LFTM. However, the algorithm is rather slow and cannot be used for large corpora (*Li et al., 2016a*).

In 2016, a spherical topic model was proposed (*Batmanghelich et al., 2016*), which is an extension of the HDP (Hierarchical Dirichlet Process) model (*Blei & Jordan, 2006*). Since the model is nonparametric, the number of topics is generated automatically. In spherical HDP, every topic (sHDP) is a von Mises-Fisher distribution in normalized embeddings' space (unit sphere). The distribution of topics in documents (proportions) are inferred with HDP. Word embeddings are learned with word2vec. The authors state that the model allows to get more coherent topics than Gaussian LDA and HDP. The implementation of this model can be found at https://github.com/Ardavans/sHDP.

*Xun et al. (2016)* use the additional information from word embeddings to get better topic models for short texts. They use word2vec embeddings trained on the Wiki dataset. Similar to Gaussian LDA model, it is assumed that every topic has a normal distribution in word embeddings space. In addition, it is supposed that every short text has only one topic. Also, the authors suppose that every word either belongs to a Gaussian topic or is one of the background words which are generated from LDA topics. An unfair coin is tossed for every word and in case of failure the word is generated from the background topic (as in LDA model). Otherwise, the word embedding is considered instead of the word and assumed that this word embedding is generated from the Gaussian topic. Inference algorithm is based on Gibbs sampling.

In *Li et al. (2016c)*, a new model called mix-vMF is proposed. The main idea is similar to the Gaussian LDA, but with von Mises-Fisher distributions used instead of normal distributions, meaning that every topic is a mix of von Mises-Fisher distributions in the space of normalized vectors on the unit sphere. The distributions of topics by documenst are Dirichlet distributions (similar to LDA), while the distributions of word embeddings by topics are used instead of the distribution of words by topics. The authors claim that von Mises-Fisher distribution can reflect the vectors' similarity in terms of cosine distance more efficiently, while a mix of such distributions is able to attribute heterogenous word embeddings to the same topic. Thus, a collection of documents is represented as a collection of word embeddings. Inference is based on Gibbs sampling. The authors use word embeddings from the GloVe model. The implementation of this model is not publicly available.

*Li et al. (2016b)* propose TopicVec model. The model creates topic embeddings for topics that implies that topic is a point in the space of word embeddings. The model takes into consideration a certain number of context words (parameter $l$) in front of every word and a topic, when assigning a probability to this word. The distribution of topics by documents is Dirichlet distribution (as in LDA). Thus, a generative model for documents is obtained. This model uses pre-trained word embeddings from PSDVec (*Li, Zhu & Miao, 2017*). The top words for every topic are found as the nearest words according to weighted cosine measure between topic embedding and word embeddings. There is a matrix of the distribution of topics by documents, however there is no matrix of the distribution of words by topics. Inference is based on variational E-M algorithm. The main advantage of the model is the rejection of the bag-of-words hypothesis and consideration of words' order. The implementation of this model can be found at https://github.com/askerlee/topicvec.

In 2017, a correlated Gaussian topic model was proposed by *Xun et al. (2017b)*. The approach is based on correlated topic model, where words are represented by their embeddings, while topics are multivariate normal distributions in the space of word embeddings. The main goal is modeling of topics and correlations between topics in the word embedding space. Word embeddings are trained separately on a large corpus with word2vec prior to topic modeling. This approach is very similar to the Gaussian LDA model with the difference that the distributions of topics for documents is log-normal, not Dirichlet. It allows to find the covariance between topics. The model inference is based on Gibbs sampling. The implementation of the model is not publicly available.

*Zhao, Du & Buntine (2017)* propose a new model, which is called WEI-FTM. This model is aimed to improve topic modeling quality for short texts. This model is a "focused topic model", meaning that each topic is focused on some set of words, which implies that the topic is a distribution not over the whole dictionary, but over its subset. Similar to LDA, each document is generated by $K$ topics, the distribution of words by topic is a Dirichlet distribution for a certain subset of words, while $\phi_{wt}$ equals zero for the rest of the words. The procedure of focus words selection is carried out as follows. A coin with success probability equal to function of the dot product between the word and the topic vector is tossed. The word is included in subset of words of the current topic in case of success, and not included in case of failure. The model is rather similar to LDA, namely, distribution of topics by documents is Dirichlet distribution and distribution of words by topics is Dirichlet distribution but only over a certain subset for every topic. Word embeddings are used only to extract focus words for every topic and are trained separately with GloVe. Model inference is based on Gibbs sampling. There are two ways to choose top words in this model. One can either use the traditional matrix $\Phi$ or consider the dot product of the word and the topic vector representation. The authors use both approaches and calculate topic coherence separately. There is no implementation of this model in the public domain.

The authors of Collaborative Language Model (CLM) (*Xun et al., 2017a*) model topics using the global context and train word embeddings using the local context simultaneously. In the framework of the model, topic embeddings are trained, and the dot product of the topic and the word embedding is used to evaluate the word's contribution into topic. The authors suggest not using pre-trained embeddings, because there are not so much open data in some fields. Thus, word embeddings use information only from a given dataset. This model is not generative and not stochastic but is solving the non-negative matrix factorization optimization task, where the target function consists of decomposition of words' co-occurrence matrix(to get word embeddings), decomposition of topic matrix (to get vector representations for topics) and norm restrictions (regularization) for every matrix. Inference is based on equalizing of target function derivatives to zero and iterative calculation of respective matrix decompositions. The authors demonstrate that their model produces coherent topics and good vector representations of words (pairs of words are sorted according to a cosine measure, then a correlation between the sorted pairs and word similarity human-made rates is calculated). The implementation of the model can be found at: https://github.com/XunGuangxu/2in1.

In *Zhao et al. (2017)*, the MetaLDA model is proposed. This model allows to use meta-information of document (*e.g.*, author or label) and words (*e.g.*, word embeddings). It is supposed that meta-information improves the quality of topic modeling for short texts. Meta-information for both words and documents is encoded in binary vectors. It is assumed that the distribution of words by topics is a Dirichlet distribution with $V$ hyperparameters (having different values for different topics), where $V$ is the size of the dictionary, and the distribution of topics by documents is a Dirichlet distribution with $T$ hyperparameters (different for every document), where $T$ is the number of topics. The binary vectors of meta-information are used to determine hyper-parameters for the Dirichlet distributions. If two words ($w_1$ and $w_2$) have similar meta-information, then the

values of hyperparameters for the distribution of topic over these words will be similar as well as their math expectations of $\Phi_{w_1,k}$ and $\Phi_{w_2,k}$. The implementation of the model can be found at https://github.com/ethanhezhao/MetaLDA/.

*Miao, Grefenstette & Blunsom (2017)* suggest several topic models based on neural variational inference (python code is available at https://github.com/zll17/Neural_Topic_Models). Neural networks is capable of approximating various functions and learning complicated non-linear distributions for unsupervised models. That is why the authors suggest using an alternative neural approach to topic modelling based on parameterized distributions over topics, which can be trained with backpropagation in the framework of neural variational inference. Neural variational inference approximates a posterior distribution of a generative model by means of variational distribution parameterized by a neural network. The authors propose three different models. The generative process of the models is as follows:

- Proportions of topics $\theta_d$ for each document $d \in D$ are distributed as $G(\mu_0, \sigma_0^2)$, where $G(\mu_0, \sigma_0^2)$ consists of neural network $\theta = g(x)$ sampling from normal distribution $x \in N(\mu_0, \sigma_0^2)$.
- Topic $z_n$ has a multinomial distribution $z_n \sim Multi(\theta_d)$ for each observed word $w_n$, n $= 1, \ldots, N_d$.
- Each word $w_n \sim Multi(\beta_{z_n})$, $n = 1, \ldots, N_d$, where $\beta_{z_n}$ is the distribution of words in topic $z_n$

Let $t \in R^{K \times L}$ be vector representations of topics and $v \in R^{V \times L}$ be vector representations of words, then distribution of words in topic k is $\beta_k = softmax(v \cdot t_k^T)$. The first proposed model is GSM (Gaussian Softmax distribution). It has the finite number of topics $K$. Proportions of topics in documents distributions are set with gaussian softmax (*Miao, Grefenstette & Blunsom, 2017*):

- $x \in N(\mu_0, \sigma_0^2)$,
- $\theta = softmax(W_1^T x)$, where $W_1$ is linear transformation.

The second proposed model is GSB (Gaussian Stick Breaking distribution). It has the finite number of topics $K$ as well. Proportions of topics in documents are set with Gaussian stick breaking process:

- $x \in N(\mu_0, \sigma_0^2)$,
- $\eta = sigmoid(W_2^T x)$ gives stick breaking proportions,
- $\theta_d = f_{SB}(\eta)$, where $f_{SB}(\eta)$ is stick breaking construction. For example, for $K = 3$, $f_{SB}(\eta_1, \eta_2) = (\eta_1, \eta_2 \cdot (1 - \eta_1), (1 - \eta_2) \cdot (1 - \eta_1))$. For $K = 4$, $f_{SB}(\eta_1, \eta_2, \eta_3) = (\eta_1, \eta_2 \cdot (1 - \eta_1), \eta_3 \cdot (1 - \eta_2) \cdot (1 - \eta_1), (1 - \eta_3) \cdot (1 - \eta_2) \cdot (1 - \eta_1))$. Thus, for any $K$, $\sum_k \theta_k d = 1$.

In the third model, the Recurrent Stick Breaking process (RSB), the number of topics is unbounded, and the distribution of topics in documents is set with the Recurrent Stick Breaking process (*Miao, Grefenstette & Blunsom, 2017*):

- $x \in N(\mu_0, \sigma_0^2)$,

- $\eta = f_{RNN}(x)$, where $f_{RNN}(x)$ is decomposed as $h_k = RNN_{SB}(h_{k-1}), \eta_k = sigmoid(h_{k-1}^T x)$. RNN denotes a recurrent neural network. Thus, the proportions for Recurrent Stick Breaking are generated sequentially by RNN.
- $\theta_d = f_{SB}(\eta)$, where $f_{SB}(\eta)$ is the same function as in the previous model.

Lower variational estimate of likelihood is used for the model inference. Variational parameters $\mu(d), \sigma(d)$ (for the document d) are generated with an inference network based on multilayer perceptron. Generative parameters (such as $t, v$ and parameters of $g(x)$) as well as variational parameters (such as $\mu(d), \sigma(d)$) are updated with stochastic gradient backpropagation algorithm. The authors demonstrate that the first two models perform better than the standard LDA model in terms of perplexity, while the third one preforms better than the HDP model.

*Bunk & Krestel (2018)* suggest a new model WELDA (Word Embedding Latent Dirichlet Allocation), which combines LDA with word embeddings. Pre-trained skip-gram word2vec embeddings are used. This model merges the classical Dirichlet distributions and multivariate normal distributions in word embeddings space. The main idea is to find vector representations for words, train standard LDA until convergence and find the parameters of normal distribution for each topic according to its top words (more precisely, according to their word embeddings). After that, additional iterations of Gibbs sampling are run, where an unfair coin is tossed for every word (with lambda probability of success, giving Bernoulli distribution) in the following way. Let word $w$ be assigned to topic $t$, then in case of success, a vector from the embeddings space of this topic is sampled (*i.e.,* from multivariate normal distribution of the topic in embeddings space), the nearest word embedding to this sampled vector is found and the corresponding word is labeled with topic $t$, and all counters are recalculated after that. Model implementation can be found at https://github.com/AmFamMLTeam/hltm_welda (not original implementation).

*Dieng, Ruiz & Blei (2020)* suggested a new topic model called embedded topic model (ETM). This model is generative and probabilistic: every document is a mixture of topics, and every word is attributed to a specific topic. At the same time, every word has a vector representation (word embedding), and topics are the vectors in the same space as well. One of the main goals of this model is enrichment of topic models with the usage of the similarity of words according to their vector representations. Let us consider this model in more detail since it is used in our experiments. Let $\rho$ denote the matrix of words' vector representations for the dictionary of a given collection. This matrix has $L \times V$ size, where $L$ is the size of vector representations of words and $V$ is the number of unique words. Each column of $\rho$ is a vector representation of a word. Also, let $\alpha_k \in \!> R^L$ denote the vector representation of topic $k$. Then, the generative process for document d can be described as follows:

1. Proportions of topics for given document, $\theta_d$, are sampled from the logit-normal distribution LN(0,I).
2. For each position $n$ in the document, a topic $z_{dn}$ is sampled from a categorical distribution $Cat(\theta_d)$ and the word is sampled according to $w_{dn} \sim softmax(\rho^T \alpha_{z_{dn}})$.

Thus, the words are generated from a categorical distribution with a parameter equal to the dot product of vector representations of the word and the topic. The matrix $\Phi$ of the distribution of words by topics can be calculated as $\phi_{vk} = softmax(\rho^T \alpha_k)|_v$. Model inference is based on maximization of the log-likelihood of documents collection. However direct computation is not possible, that is why variational inference is used. A family of additional multivariate normal distributions is used in inference, whose parameters (vector of means and covariance matrix) are evaluated with a neural network, which takes documents as inputs and outputs the parameters $(\mu_d, \Sigma_d)$ for every document. Thus, the evaluation of the parameters of auxiliary distributions in variational inference is carried out by a special neural network. Inference algorithm can be described as follows:

1. Initialize model and variational parameters $v_\mu, v_\Sigma$
2. Iterative steps:
   (a) Compute $\phi_{\cdot k} = softmax(\rho^T alpha_k)$ for every topic k
   (b) Choose a minibatch of documents (B)
   (c) For every document $d \in B$:

   - Construct normalized bag-of-words representation of document $(x_d)$
   - Compute $\mu_d = NN(x_d, v_\mu)$
   - Compute $\Sigma_d = NN(x_d, v_\Sigma)$
   - Sample $\theta_d \sim LN(\mu_d, \Sigma_d)$
   - For each word $w_{dn}$ in the document $d$: Compute $p(w_{dn}|\theta_d) = \theta_d^T \phi_{w_{dn}}$

   (d) Compute variational lower bound (ELBO) and its gradient
   (e) Update values of $\alpha_{1:K}$
   (f) Update values of variational parameters $v_\mu, v_\Sigma$

The authors propose two options for the model: (1) with pre-trained embeddings; (2) learning the embeddings as part of the fitting procedure. The numerical experiments of the authors demonstrate that the model with pre-trained embeddings gives slightly better quality on average in terms of topics' interpretability and predictive ability compared to the alternative. Also, ETM significantly improves the quality measures such as semantic coherence and predictive ability in comparison to LDA. It should be noted, that skip-gramm pre-trained embeddings are used in this work; however, the authors admit the possibility to use other types of word embeddings. Also, the number of topics is set manually in this model leaving the problem of selecting the number of topics open.

Further development of ETM model was proposed in *Harandizadeh, Priniski & Morstatter (2022)*. The new proposed model (keywords assisted ETM) incorporates user knowledge in the form of informative topic-level priors over the vocabulary. Namely, the user specifies a set of seed word lists associated with topics of interest, that, in turn, guides statistical inference.

At the end of 2019, the W-LDA model was proposed (*Nan et al., 2019*). This is a neural topic model based on Wasserstein autoencoder with Dirichlet prior on the latent document-topic vectors. The encoder consists of a multi-layer perceptron mapping bag-of-words representation of a document to an output layer of K units, then softmax is applied to obtain the document-topic vector $\theta$. Given $\theta$, the decoder consists of a single layer

neural network mapping $\theta$ to an output layer of V units, then softmax is applied to obtain a probability distribution over the words in the vocabulary ($\hat{w}$). Thus, $\hat{w}_i = \frac{exp(h_i)}{\sum_{j=1}^{V} exp(h_j)}$, where $h = \beta\theta + b$, $\beta$ is the matrix of topic word vectors as in LDA and $b$ is an offset vector. The top words of each topic can be extracted based on the decoder matrix weights (*i.e.,* top entries of $\beta_k$ sorted in descending order). The authors demonstrate that their model produces significantly better topics compared to LDA, ProdLDA (*Tolstikhin et al., 2018*), NTM-R (*Ding, Nallapati & Xiang, 2018*).

WTM-GMM is an improved version of the original W-LDA https://zll17.github. io/2020/11/17/Introduction-to-Neural-Topic-Models#WTM-MMD. Gaussian mixture distribution is considered as prior distribution. The authors propose two types of evolution strategy: gmm-std and gmm-ctm. The gmm-std adopts Gaussian mixture distribution, whose components have fixed means and variances. In contrast, the components of Gassuian mixture distribution of the gmm-ctm are adjusted to fit the latent vectors through the whole training process. The number of the components is usually set as the number of topics. Empirically, the WTM-GMM model usually achieves better performance than W-LDA in terms of topic coherence.

In the spring of 2020, the Bidirectional Adversarial Topic model (BAT) was proposed (*Wang et al., 2020*). This model uses a Dirichlet distribution as a prior distribution of topics (analogous to LDA). Moreover, an extension of this model, Gauusian BAT, which is able to account the words similarity based on their vector representations, was also proposed. In this work, bidirectional adversarial training is used for the first time in topic modelling. The BAT and Gaussian-BAT models use a Dirichlet prior for modelling topics. Let $V$ be the size of the dictionary and $K$ be the number of topics. BAT model consists of 3 components: (1) The encoder which takes a V-dimensional document representation ($\vec{d_r}$) as input and transforms it into a K-dimensional distribution of topics in the document ($\vec{\theta_r}$); (2) The generator takes a random distribution of topics in the document as input ($\vec{\theta_f}$), which is sampled from the Dirichlet distribution, and generates an artificial V-dimensional distribution of words ($\vec{d_f}$). (3) The discriminator takes a real pair of distributions $\vec{p_r} = [\vec{\theta_r}, \vec{d_r}]$ and an artificial pair of distributions $\vec{p_f} = [\vec{\theta_f}, \vec{d_f}]$ as input. After that, it has to distinguish real distributions and the artificial ones. The output of discriminator is used to train the encoder, generator and discriminator in adversarial training. The encoder contains a V-dimensional layer of distribution of words in a document, an S-dimensional layer of representations and a K-dimensional layer of distribution of topics in a document. Each document $d$ has its own representation ($\vec{d_r}$) weighted with TF-IDF. Firstly, the encoder projects $\vec{d_r}$ into S-dimensional semantic space by means of the representation layer as follows: $\vec{h_s^e} = BN(W_s^e \vec{d_r} + \vec{b_s^e})$, $\vec{o_s^e} = max(\vec{h_s^e}, leak * \vec{h_s^e})$, where $W_s^e \in R^{S \times K}$ is the weight matrix of the representation layer, $b_s^e$ is the bias, $h_s^e$ is the state vector normalized with batch normalization, leak denotes the parameter of Leaky ReLu activation function, $o_s^e$ is the output of the representation layer. Then, the encoder converts $o_s^e$ onto a K dimensional topic space: $\vec{\theta_r} = softmax(W_t^e \vec{o_s^e} + \vec{b_t^e})$, where $W_t^e \in R^{K \times S}$ is the weight matrix of the topic distribution layer, $\vec{b_t^e}$ is the bias of this layer, $\vec{\theta_r}$ denotes the topic distribution of document $\vec{d_r}$, while $\theta_r^k$ is the proportion of topic $k$ in this document. The generator

projects the distribution of topics in documents onto the distribution of words in the document in contrast to the encoder. Thus, the generator consists of a K-dimensional layer of the distribution of topics in documents, an S-dimensional representation layer and a V-dimensional layer of the distribution of words in a document. The distribution of topics in documents $\overrightarrow{\theta_f}$ ($\theta_f^k$ denotes the proportion of topic k in the document) is assumed to be Dirichlet distribution with a K-dimensional parameter $\overrightarrow{\alpha}$. First, the generator projects the distribution of topics in documents onto an S-dimensional representation space: $\overrightarrow{h_s^g} = BN(W_s^g \overrightarrow{\theta_f} + \overrightarrow{b_s^g})$, $\overrightarrow{o_s^g} = max(\overrightarrow{h_s^g}, leak * \overrightarrow{h_s^g})$, where $W_s^g \in R^{S \times \times K}$ is the weight matrix of the representation layer, $\overrightarrow{b_s^g}$ is the bias, $\overrightarrow{h_s^g}$ is the state vector normalized with batch normalization, leak denotes the parameter of Leaky ReLu activation function, $\overrightarrow{o_s^g}$ is the output of the representation layer. Then $\overrightarrow{o_s^g}$ is transformed into the distribution of words in a document by means of linear layer and softmax: $\overrightarrow{d_f} = softmax(W_w^g \overrightarrow{o_s^g} + \overrightarrow{b_w^g})$, where $W_w^g \in R^{V \times S}$ is the weight matrix of the distribution of words, $\overrightarrow{b_w^g}$ is the bias of the layer, $\overrightarrow{d_f}$ denotes the distribution of words corresponding to $\overrightarrow{\theta_f}$. For each word $v$, $d_f^v$ is the probability of this word in the artificial document $\overrightarrow{d_f}$. The discriminator consists of the 3 layers: a $V + K$-dimensional layer for joint distribution, an S-dimensional layer of representations and an output layer. The main task of the discriminator is to distinguish real input data $\overrightarrow{p_r} = [\overrightarrow{\theta_r}; \overrightarrow{d_r}]]$ from artificial ones $\overrightarrow{p_f} = [\overrightarrow{\theta_f}; \overrightarrow{d_f}]$. The output of the discriminator is $D_{out}$. Large $D_{out}$ value indicates that the generator defines the input as real. The authors also propose a modified Gaussian-BAT with a modified generator to consider information about the relationship of words based on vector representations of words. The multivariate normal distribution ($N(\overrightarrow{\mu_k}, \Sigma_k)$) is used to model topic $k$, where $\overrightarrow{\mu_k}$ and $\Sigma_k$ are the parameters learned during training. The probability of every word $v$ is calculated as follows: $p(\overrightarrow{e_v}|topic = k) = N(\overrightarrow{e_v}; \overrightarrow{\mu_k}, \Sigma_k)$, $\phi_{vk} = \frac{p(\overrightarrow{e_v}|topic=k)}{\sum_{v=1}^{V}} p(\overrightarrow{e_v}|topic = k)$, where $\overrightarrow{e_v}$ denotes the vector representation of the word $v$, $\phi_{\cdot k}$ is the normalized distribution of words in the k-th topic. The artificial distribution $\overrightarrow{d_f}$ corresponding to a certain $\overrightarrow{\theta_f}$ can be computed as follows: $\overrightarrow{d_f} = \sum_{k=1}^{K} \overrightarrow{\phi_k} * \theta_f^k$. The encoder and decoder are the same as in BAT. Pairs of real distributions $\overrightarrow{p_r} = [\overrightarrow{\theta_r}; \overrightarrow{d_r}]]$ and pairs of artificial distributions $\overrightarrow{p_f} = [\overrightarrow{\theta_f}; \overrightarrow{d_f}]$ are considered as random samples from two (K+V) - dimensional joint distributions $P_r$ and $P_f$. The main task is to make $P_f$ as close to $P_r$ as possible. Wasserstein distance is used as the proximity measure between $P_r$ and $P_f$. A detailed algorithm of training is given in *Wang et al. (2020)*. The authors demonstrate that the proposed models perform better than the standard LDA and GSM models in terms of the 'topic coherence' measure.

In *Xu et al. (2022)*, a neural topic model with deep mutual information estimation (NTM-DMIE) is proposed. This method maximizes the mutual information between the input documents and their latent topic representation. The framework of NTM-DMIE consists of two main components, namely, Document-Topic Encoder and Topic-Word Decoder. The Document-Topic Encoder simulates the document-topic distribution as in LDA and learns topic representations of documents. Moreover, in the encoder, mutual information is estimated between the documents and their topic representations. The Topic-Word Decoder learns the topic-word distribution as in LDA. The authors

demonstrate that the proposed model outperforms state-of-the-art neural topic models in terms of 'topic coherence' and 'topic uniqueness' metrics.

In *Shao et al. (2022)*, the role of embeddings and their changes in embedding-based neural topic models is studied. Moreover, the authors propose an embedding regularized neural topic model (ERNTM), which applies the specially designed training constraints on word embeddings and topic embeddings to reduce the optimization space of parameters. The authors compare the proposed model with the baseline models, such as ETM, GSM, NTM (*Ding, Nallapati & Xiang, 2018*) and demonstrate its competitiveness.

To increase the quality of topic modeling on short texts, a neural topic model integrating SBERT and data augmentation was proposed in *Cheng et al. (2023)*. The authors introduce a data augmentation technique that uses random replacements, insertions, deletions, and other operations to increase the robustness of text data and incorporate it with keyword information obtained through the TextRank algorithm. Then, the augmented text data is vectorized and used as input for a BiLSTM-Att module to obtain long-distance dependency information and overcome the influence of noisy words. The authors also propose SBERT model, which, in contrast to BERT, takes the entire sentence as a processing unit. The information that was enhanced through data augmentation and processed through the attention mechanism is merged with semantic feature information. The resulting feature information is fed into a neural topic model based on ProdLDA model.

In general, topic modeling with the application of neural networks is actively developing. One of the best reviews in this area is *Zhao et al. (2021)*, although this work is from 2021. The active development of transformer models has given rise to several new works using topic modeling as an auxiliary tool. For example, in *Giang, Song & Jo (2022)*, topic modeling was used to improve the segmentation of high-level images as follows (TopicFM model). This approach represents an image as a set of topics marked with different colors, *i.e.,* encodes high-level contextual information of images based on the topic modeling strategy in data analysis. Each topic is an embedding fed into the 'cross-attention layer' input, characterized by three matrices (queries, keys, values). Thus, a standard transformer scheme is used. At the transformer's output, probabilities characterizing the distance between the attribute $F_i$ and separate topics $T$ are obtained. Further, one can obtain a similar matrix for different images and then compare the images with each other. Thus, TopicFM provides reliable and accurate feature-matching results even in complex scenes with large changes in scale and viewpoint.

In *Wang et al. (2023)*, large language models (LLMs) are considered implicit topic models. The authors of this paper, relying on the fact that LLMs are Monte Carlo generative models, propose to consider the generation process in dependence on the topic and the topic-token matrix. With this point of view, topic modeling is reduced to a classification procedure, that is, to obtaining a label for each document in the form of the probability that this document belongs to a particular topic. This paper analyzed the following LLMs: GPT2, GPT3, GPTS-instruct, GPT-J, OPT, and LLaMA. In their tests, the authors proposed a two-stage algorithm that first extracts latent conceptual lexemes from a large language model and then selects demonstrations from clues that are most likely to predict the corresponding conceptual lexemes.

The Zero Shot Classification technology (*Brown et al., 2020*) should also be noted. In the framework of this technology, large language models are used to classify text data using transfer learning. That is, the LLM-based classifier can output the probabilities that the text belongs to different topics, and the user specifies the topics. Thus, it is possible to construct a simple text clustering algorithm for a given set of topics. Further development of this direction is demonstrated in *Ding et al. (2022)*. In this paper, the authors propose a topic classification system originally trained on Wikipedia. Thus, the trained classifier can classify an external document on various topics with high accuracy. The proposed framework is also based on zero-shot classification technology.

We would also like to mention some other applications in the field of NLP where topic modeling is used as an auxiliary tool. *Joshi et al. (2023)* proposes the 'DeepSumm' method for text summarization. In the framework of this approach, each sentence is encoded with two different recurrent neural networks based on the probability distributions of the topics and embeddings. Then a sequence-to-sequence network is applied to the encoding of each sentence. The outputs of the encoder and decoder in the sequence-to-sequence networks are combined after weighting by an attention mechanism and converted into an estimate by a multilayer perceptron network. Accordingly, several scores are obtained for each sentence, namely the score obtained using the Sentence Topic Score (STS) topic model, and the score obtained using embeddings: Sentence Content Score (SCS). In addition, the authors offer Sentence Novelty Score (SNS) and Sentence Position Score (SPS). Based on these four scores, a Final Sentence Score (FSS) is calculated. Accordingly, all sentences are ranked according to the final score, and a brief summary of the text is the set of sentences that received the maximum value.

In conclusion, we would like to note that in the last two years, there has been a change in focus from the development of topic models to the use of models in conjunction with large language models, or even to the replacement of topic sampling procedures for classification with style transfer, that is, the widespread use of zero-shot technology. At the same time, embeddings have become an integral part of transformers when working with various NLP tasks.

## APPENDIX B

In this section, we briefly describe the main models of word embeddings.

**Word2vec model**
The technology of word embeddings is based on the hypothesis of local co-occurrence of words. In the framework of this hypothesis, it is assumed that words that often occur with similar surrounding words have the same or similar semantic meaning. This hypothesis was proposed by *Mikolov et al. (2013)*. The probability of occurrence of word $w_0$ with surrounding words is expressed as follows: $p(w_0|W_c) = \frac{\exp(s(w_0, w_c))}{\sum_{w \in V} \exp(s(w_i, w_c))}$, where $w_0$ is a vector representation of the target word, $w_c$ is the context vector, $s(w_0, w_c)$ is a function matching two vectors with a number, for example, a distance measure such as a cosine measure.

CBOW (continuous bag of words) is a model where the network is trained on a continuous sequence of words. In the framework of this model, the order of words is not important. In this model, a sequence of $2k+1$ words is used, where the central word is is the word under study, and the context vector is build based on the corresponding surrounding words. Thus, each vector has a set of words, which often occur together. In the CBOW model, the probability of words is based on minimization of Kullback–Leibler divergence: $KLB(p||q) = \sum_{x \in V} p(x) \ln(\frac{p(x)}{q(x)})$, where $p(x)$ is the probability distribution of words from the dataset, $q(x)$ is the word distribution generated by the model. Skip-gram model is a model of phrases with a gap. This model is similar to the previous model. The principle of the CBOW model is a prediction of a word given context, and the principle of skip-gram model is a prediction of context given a word. In general, the word2vec model is optimized based on negative sampling procedure (*Mikolov et al., 2013*). Negative sampling is a way to create negative examples for model learning (i.e, to show pairs of words that are not neighboring in the context).

**GloVe model**

The main idea of the GloVe model is to extract semantic relations between words using the matrix of co-occurrence of words. This model minimizes the difference between the product of word vectors and the logarithm of the probability of their co-occurrence using stochastic gradient descent (*Pennington, Socher & Manning, 2014*). In this case, it is possible to connect satellites of one planet or the city's postal code with its name, that could not be done using the Word2vec model.

**FastText model**

The FastText model is an extension of the Word2vec model. In the FastText model, skip-gram, negative sampling, and model of symbolic n-grams are used (*Joulin et al., 2017*). Each word is presented as a composition of several sequences of symbols of a certain length. In this approach, word embedding is the sum of these n-grams. Parts of words are also likely to occur in other words, making it possible to produce vector representations for rare words.

**Doc2vec model**

The Doc2vec model allows one to map an entire document into a numeric vector (*Le & Mikolov, 2014*). The developers of the concept proposed the following algorithm. Each paragraph is represented as vector of words, where each word is represented as a numeric vector, characterizing the proximity of words to each word in a paragraph. Paragraph vectors and vectors of words are averaged to improve word prediction. In general, this approach is analogous to the word2vec model, with the only difference being that the window slides over the document's paragraphs. Moreover, in the framework of this model, the algorithm of the stochastic gradient is used for the optimization of the softmax function.

### ElMo model

In *Peters et al. (2018)*, the authors proposed an approach where word vectors are trainable functions of inner states of a deep bidirectional language model (biLM), which was previously trained on a large corpus of texts. The authors used deep network 'bidirectional LSTM' with several normalization layers. This approach uses a character-by-character data representation strategy, so ElMo (Embeddings from Language Models) provides three layers of representations for each input token, including those that are outside the training set due to pure character input. In contrast to this approach, traditional word embedding methods only provide one level of representation for lexemes from a fixed vocabulary.

Further development of algorithms for building embeddings went through using large language models; namely, embeddings began to be used as an input to transformers that deal with various NLP tasks, such as translation, text summarization, sentiment analysis, and others. In addition, the architecture of transformers began to be used to build embeddings.

### Tuning of embeddings for 'transfer learning' tasks

In *Cer et al. (2018)*, the authors propose two models for encoding sentences into embeddings, which are specifically aimed at the transfer of learning. The proposed variants of coding models make it possible to find a compromise between accuracy and computational costs since training large language models is an extremely costly procedure, both in terms of time and finances. The first model encodes sentences into embeddings based on the 'sub-graph of the transformer' architecture (*Vaswani et al., 2017*). This sub-graph uses attention to calculate context-sensitive representations of words in a sentence taking into account the order and the identity of all other words. The context-aware word representations are converted to a fixed-length sentence encoding vector by computing the element-wise sum of the representations at each word position. The encoder takes as input a string with PTB (Penn Treebank tokenization) tokens in lowercase and outputs a 512-dimensional vector as the sentence embedding.

The second encoding model is based on the Deep Averaging Network (DAN) (*Iyyer et al., 2015*), in which the embeddings for words and bigrams are first averaged together and then passed through a feedforward deep neural network (DNN) to obtain the final sentence embeddings. Like the Transformer encoder, the DAN encoder takes a lowercase PTB token string as input and produces a 512-dimensional sentence embedding. The authors have shown that transfer learning based on the transformer-based embedding encoder performs as well or better than learning based on the DAN encoder. Models in 'transfer learning' tasks that use sentence-level embeddings tend to perform better than models that only use word-level transfers.

### Electra model

In *Clark et al. (2020)*, a new method for pretraining text encoders based on discriminators is proposed. The proposed model (Electra: Efficiently Learning an Encoder that Classifies Token Replacements Accurately) consists of two generator networks and a discriminator (based on transformers). The idea of such a network is as follows: the learning mode based on masking words is replaced by a masking model of lexemes taken from the generator

network. Then, instead of training a model that predicts the original identity of the latent lexemes, a discriminative model is trained that predicts whether each token in the latent form has been replaced by a generator pattern or not. As the authors have shown in their experiments, this architecture is superior to models such as BERT and GPT-2 and is comparable to the RoBERTa and XLNet models with a lower amount of training data. In addition, this paper shows the procedure for generating embeddings based on token masking.

**Acoustic word embeddings**

Recently, works that construct embeddings not from text data but from images and audio tracks have appeared. For example, in *Jacobs & Kamper (2023)*, an algorithm for constructing acoustic embeddings (AWE) is proposed, which are speech segments that encode phonetic content so that different implementations of the same phonetic content have the same embeddings. This model is based on a recurrent network for matching word segments in embedding.

**Recent applications of word embeddings**

In general, the works of 2022-mid-2023 are not focused on developing new embedding models but on creating new architectures for large language models, as well as on forming various ways to use embeddings. For example, in *Muennighoff (2022)*, the author considers the SGPT (decoder-only transformers) model for semantic search and extraction of meaningful embeddings based on prompt engineering and fine-tuning. In addition, websites are being actively developed where users can either find ready-made embeddings, as was done in the framework of this work, or sites that contain ready pre-trained neural networks tailored for various NLP tasks, including building embeddings. One such popular resource is the 'Hugging Face' repository (https://huggingface.co/blog/getting-started-with-embeddings). The most recent review (April 2023) of large language models is given in *Yang et al. (2023)*. This paper discusses the areas of application of transformers such as decoder-only, encoder-only, and encoder–decoder architectures in the context of various NLP tasks. It should be noted that, by definition, the architecture of the Transformers type uses an embedding layer as an input layer. Hence, the main flow of scientific work in 2022–2023 is related to developing different architectures that use the above embedding construction schemes.

In addition, we would like to mention *Wang et al. (2019)*, in which the authors consider the most popular models for building embeddings, such as Continuous-Bag-of-Words (CBOW), Skip-Gram, a model based on Co-occurrence Matrix, FastText, N-gram Model, Deep Contextualized Model, and other obsolete dictionary-based models. This work is notable for the fact that the authors try to build quality metrics for embedding construction models, regardless of the context of the NLP task. The authors of this paper have formulated several characteristics that embedding algorithms must comply with. For example, the test data on which it is recommended to test embedding construction models should be varied with a good spread in the word space. Common and rare words should be included in the estimation. The performance of embedding models should also have good statistical

significance in order to be able to rank such models. The authors conducted many interesting and useful experiments for the end users, in which they showed how the nature of embeddings changes when they are built for such NLP tasks as (1) part-of-speech (POS) tagging, (2) named entity recognition, (3) sentiment analysis, (4) neural machine translation (NMT). The result of this work is a guide to the selection of appropriate evaluation methods for various applications. The authors showed that there are many factors that affect the quality of embeddings. In addition, the authors pointed out that, until now, there are no ideal methods for evaluating testing a subspace of words for the presence of linguistic relationships, since it is difficult to understand exactly how embeddings encode linguistic relationships.

### Funding

The results of the project "Modeling the structure and socio-psychological factors of news perception", carried out within the framework of the Basic Research Program at the National Research University Higher School of Economics (HSE University) in 2022, are presented in this work. The funders had no role in study design, data collection and analysis, decision to publish, or preparation of the manuscript.

### Grant Disclosures

The following grant information was disclosed by the authors:
"Modeling the structure and socio-psychological factors of news perception".
The Basic Research Program at the National Research University Higher School of Economics (HSE University) in 2022.

### Competing Interests

The authors declare there are no competing interests.

### Author Contributions

- Sergei Koltcov conceived and designed the experiments, analyzed the data, authored or reviewed drafts of the article, and approved the final draft.
- Anton Surkov conceived and designed the experiments, performed the experiments, analyzed the data, performed the computation work, prepared figures and/or tables, authored or reviewed drafts of the article, and approved the final draft.
- Vladimir Filippov conceived and designed the experiments, performed the experiments, performed the computation work, prepared figures and/or tables, and approved the final draft.
- Vera Ignatenko analyzed the data, prepared figures and/or tables, authored or reviewed drafts of the article, and approved the final draft.

### Data Availability

The datasets are available in Zenodo: Ignatenko Vera. (2023). Topic modeling datasets [Data set]. Zenodo. https://doi.org/10.5281/zenodo.8407610.

The source codes are available in Zenodo: skoltsov. (2023). veraignatenko/TM-with-neural-networks: For Zenodo (v1.0). Zenodo. https://doi.org/10.5281/zenodo.8410811.

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
