# Peer review of "Topic models with elements of neural networks: investigation of stability, coherence, and determining the optimal number of topics"

_PeerJ Computer Science, doi:10.7717/peerj-cs.1758_

## Round 0.1 · original submission · Major Revisions

The reviewers raised a number of problems that need to be addressed in a new version of the manuscript.

Reviewer 1 ·

Basic reporting

The paper has some shortcomings:

(1) The maniuscript text is often vague and long-winded. Please proofread the article and make it more clear and continues.
(2) In several instances regarding word embedding, I also suggested to provide more relevant discussion and recent literature.
(3) Please recent literature references ( 2022-2023).

Experimental design

The paper has following shortcomings in regards to mthodology and data analyses:
(1) Proposed and existing models are not decsribe in tabular and graphical form.
(2) It looks proposed model is computationally expensive than existing models.

Validity of the findings

It looks proposed model is computationally expensive than existing models, therefore it is recommended that:
(1) At least one more dataset should be utilized to its full extent for validity of the proposed scheme.
(2) Please provide the quantitative comparison of proposed work with the existing work.

Reviewer 2 ·

Basic reporting

a) The language of this article is professional, clear and technical.
b) The introduction and background part have elaborated the gap and problem this work aiming to solve. Related literatures or prior works have been reviewed in introduction part.
c) The structure of this article is identical to the requirement of acceptable format. Figures in this article also support the corresponding content. However, in my option, all figures should be pictured by Orgin.
d) All the results in this article and supplementary materials can support the hypotheses or problem definition in this article.
e) Symbol definitions are clear and relevant.

Experimental design

a) This article is in accordance with the Aims and Scope of the journal.
b) Question definition and gap between prior works and this article is clear and the following content also discuss the pathway to fill this gap.
c) The experimental design basically conforms the ethical standard.
d) Methods introduction or discussion is clear.

Validity of the findings

a) The novelty of this article is acceptable. It really solves the problem mentioned in introduction and this problem is actually meaningful in the real world. Results data and discussion are sufficient.
b) The conclusion part is well stated and summarize the whole content in the article.

Additional comments

Generally speaking, it is an interesting work. The problem it solves seems to be minor but actually improve the related researches. By reviewing this paper, I think authors devote themselves to solve this problem step by step. However, all the figures in this paper need to be repictured.

---

## Round 0.2 · accepted · Accept

The reviewer positively assessed the article and therefore I can recommend it for acceptance and publication.

Reviewer 1 ·

Basic reporting

Thanks for all the corrections. I am fine with the extent of the changes you have made to the manuscript and have properly responded to the comments from the previous review. Overall, the quality of the article has increased. However, I suggest listing Github URLs to model implementations such as https://github.com/ethanhezhao/MetaLDA/, https://github.com/AmFamMLTeam/hltm_welda, etc. as references instead of having them in text.

Experimental design

No Comments

Validity of the findings

No Comments

Additional comments

The authors incorporated all suggestions. Thus, I suggest to accept the article for publication.